# Learning from A Single Markovian Trajectory: Optimality and Variance Reduction

**Zhenyu Sun**,* **Ermin Wei**\*†

\*Department of Electrical & Computer Engineering, Northwestern University
†Department of Industrial Engineering & Management Sciences, Northwestern University
{zhenyu.sun, ermin.wei}@northwestern.edu

## Abstract

In this paper, we consider the general stochastic non-convex optimization problem when the sampling process follows a Markov chain. This problem exhibits its significance in capturing many real-world applications, ranging from asynchronous distributed learning to reinforcement learning. In particular, we consider the worst case where one has no prior knowledge and control of the Markov chain, meaning multiple trajectories cannot be simulated but only a single trajectory is available for algorithm design. We first provide algorithm-independent lower bounds with $\Omega(\epsilon^{-3})$ (and $\Omega(\epsilon^{-4})$) samples, when objectives are (mean-squared) smooth, for any first-order methods accessing bounded variance gradient oracles to achieve $\epsilon$-approximate critical solutions of original problems. Then, we propose **Ma**rkov-**C**hain **SPIDER** (MaC-SPIDER), which leverages variance-reduced techniques, to achieve a $\mathcal{O}(\epsilon^{-3})$ upper bound for mean-squared smooth objective functions. To the best of our knowledge, MaC-SPIDER is the first to achieve $\mathcal{O}(\epsilon^{-3})$ complexity when sampling from a single Markovian trajectory. And our proposed lower bound concludes its (near) optimality.

## 1 Introduction

Many modern learning tasks can be modeled as stochastic optimization problems. Specific applications range from (un)supervised learning, reinforcement learning to large-scale generative models. In particular, the surge in deep learning and large models has made first-order methods—those that rely solely on gradient information—attractive due to their ease of implementation and computational efficiency [1, 40].

Yet, the commonly assumed i.i.d. data setting rarely reflects real-world scenarios. In many practical applications, data arrive sequentially and exhibit temporal dependencies, often shaped by underlying dynamical systems that can be modeled as Markov processes. For instance, in reinforcement learning, data are collected through interactions with an environment governed by a Markov decision process (MDP), producing inherently correlated samples [35]. In recommendation systems, user feedback unfolds over time, where each action depends on previous interactions in a Markovian fashion [2]. Time-series data—from sensors to financial markets—are also naturally Markovian [8, 13]. MCMC methods, widely used in Bayesian inference, rely on Markov chains to sample from complex posteriors [9, 28]. Even in language modeling, transformer architectures produce token sequences with local dependencies that can be interpreted through a Markovian lens [1, 27].

These examples point to a clear fact: i.i.d. assumptions are sometimes insufficient, and optimizing in the presence of Markovian data requires more careful analysis. As a result, a growing body of work has emerged to study stochastic optimization under Markov sampling, spanning areas such as reinforcement learning [7, 39], distributed optimization [33, 15], and federated learning [34]. These

39th Conference on Neural Information Processing Systems (NeurIPS 2025).

developments underscore the need for a more unified and general theoretical foundation that can account for the challenges introduced by Markovian dynamics.

Compared to the extensively studied i.i.d. setting, analyzing first-order optimization methods under Markov sampling introduces new and significant challenges due to data correlation. In the i.i.d. case, each data point is sampled independently from a fixed distribution, yielding unbiased estimates of the stochastic gradient. This independence enables the direct application of standard tools such as variance bounds and concentration inequalities [29, 22, 25]. However, when data are generated by a Markov process, consecutive samples are inherently correlated, complicating the statistical properties of gradient estimates. Specifically, the Markovian dependence introduces bias into the gradient estimates, which disrupts the foundational assumptions that are crucial for i.i.d.-based analysis. As a result, many classical techniques no longer apply directly, and alternative tools are required to handle the joint effects of bias and temporal correlation [7, 14, 31, 33, 21].

There are fruitful results about stochastic optimization with Markovian samples in literature, where a large amount focus on designing new algorithms and deriving corresponding sample complexity upper bounds. A common assumption placed by these works is that the algorithm is implemented in the simulated environment, meaning multiple trajectories can be incorporated to facilitate the performance of the algorithm. For example, in [36, 23] multiple trajectories are sampled from the underlying Markov chain for an update of policy in the RL problem. And variance-reduced techniques are incorporated by utilizing multiple trajectories to obtain better convergence rates and sample complexities [30, 26, 37, 38]. However, in practice one may have no access to multiple trajectories due to no prior knowledge and uncontrollability of the Markov chain. In such case, once the sampling begins, the chain keeps evolving and cannot be restarted, i.e., only a single trajectory is available for algorithm design. Based on this restriction, [14, 11] establish SGD-based approaches with batched samples drawn from a single trajectory and [6] further proposes the accelerating versions and generalizes them to variational inequalities. However, these methods suffer from slow convergence rates or high sample complexity, which is mainly because no variance reduction technique is considered as the i.i.d. counterpart. Moreover, when considering variance reduction for the Markov setting, biasness and time dependence make the analysis developed for the i.i.d. case fail to hold, which bring challenges in the derivation of new mathematical tools. Therefore, in this paper we are interested in understanding whether variance reduction can be achieved in the Markov sampling case when only one trajectory is available and aim to derive a new variance-reduced method to reduce sample complexity. Besides, we would also like to figure out what is the fundamental limit of our method and all existing methods. To achieve this, we further develop lower bound of sample complexity under both smooth and mean-squared smooth (which is commonly assumed for variance reduction applications) settings for any first-order algorithms, which nearly match the upper bound of our method. This concludes the near min-max optimality of our algorithm. Our main contributions are summarized as follows:

- We provide the algorithm-independent sample complexity lower bound for any first-order methods of stochastic non-convex optimization problems, given data samples are generated by a Markov chain for smooth functions. Our lower bound shows a complexity with the order of $\epsilon^{-4}$, which nearly matches the upper bound of algorithm MAG provided in [11, 6].

- We also consider the case of mean-squared smooth functions, which shows broad applications in variance reduction for the i.i.d. case. We provide a lower bound of order $\epsilon^{-3}$ when samples are Markovian.

- We then propose a new algorithm, called MaC-SPIDER. The convergence and sample complexity analysis is provided for MaC-SPIDER, which indicates nearly the same order of $\epsilon^{-3}$, hence combining with our lower bound demonstrates its (near) min-max optimality.

**Variance reduction for non-convex stochastic optimization.** Variance reduction has drawn much attention very recently. It effectively controls the variance to a low level and hence improves sample efficiency of solving stochastic optimization problems. Mature variance-reduced techniques have been successfully applied to the i.i.d. case to reduce sample complexity. Popular methods include but not limited to SAG [32], SVRG [20, 4], SPIDER [16]. In particular, it is shown that SPIDER can achieve the optimal order of $\epsilon^{-3}$ sample complexity. However, in practice data might not be i.i.d. but Markovian. For the RL problem, variance-reduced methods have been developed by assuming multiple trajectories can be used for every iteration of the algorithm. Under this assumption,

incorporating with SVRG gives a sample complexity of $\mathcal{O}(\epsilon^{-10/3})$ [30, 37] and $\mathcal{O}(\epsilon^{-3})$ is achievable when combining with SPIDER [38].

**Lower bounds for stochastic optimization.** The sample complexity of first-order stochastic optimization has been a central topic of study over the past two decades. Among these developments, lower bounds play a fundamental role, as they characterize the inherent difficulty of optimization and set performance limits for all algorithms within a given class. In the full-batch setting, where the algorithm has access to all data, a lower bound of $\Omega(\epsilon^{-2})$ has been established for smooth non-convex objectives [10], and this rate is achieved by standard Gradient Descent [18]. Under the i.i.d. sampling assumption, lower bounds of $\Omega(\epsilon^{-2})$ for convex objectives [3] and $\Omega(\epsilon^{-4})$ for general non-convex objectives [5] have been proven. The latter improves to $\Omega(\epsilon^{-3})$ under a mean-square smoothness assumption [5]. Correspondingly, stochastic gradient descent (SGD) [17, 18] and SPIDER [16] achieves matching upper bounds in both settings, respectively. In the Markovian sampling regime, where data are temporally correlated, complexity results explicitly depend on the chain's mixing or hitting time, denoted by $\tau$. For strongly convex functions, [6] establishes a lower bound of $\Omega(\tau \log(\epsilon^{-1}))$, while for general convex objectives, [12] proves a lower bound of $\Omega(\tau \epsilon^{-2})$, which matches the known upper bound for SGD in that setting. For non-convex objectives, [14] recently derived a loose lower bound of $\Omega(\tau \epsilon^{-1})$, whereas the best-known upper bound remains $\tilde{\mathcal{O}}(\tau \epsilon^{-4})$.

## 2 Problem Formulation

Consider the general stochastic optimization problem,

$$\min_x \quad F(x) := \mathbb{E}_{s \sim \pi}[f(x; s)] \tag{1}$$

where $s \in \mathcal{S}$ for $\mathcal{S}$ being the support, and $\pi$ denotes some unknown underlying distribution. In this paper we focus on the Markovian case, i.e., we assume that the samples $\{s_t\}_{t=0}^{\infty}$ form a sequence generated by some underlying Markov chain with its stationary distribution being $\pi$. Moreover we focus on countable-state Markov chains, meaning the state spaces are countable but may not be finite. Note that the Markovian setting reduces to the i.i.d. setting by decoupling the dependence across time.

Since exactly solving (1) is NP-hard [19], by restricting to first-order methods, we search for an $\epsilon$-approximate critical solution, which is widely adopted by literature [10, 5, 6] of $F(x)$ defined in (1). In particular, given differentiable function $F : \mathbb{R}^d \to \mathbb{R}$, our goal is to find some $x$ such that

$$\|\nabla F(x)\| \le \epsilon$$

for any $\epsilon > 0$.

### 2.1 Function class

Particularly, we consider all smooth functions in the following class:

$$\mathcal{F}(\Delta, L) := \big\{ F : \mathbb{R}^d \to \mathbb{R} \mid F(0) - \inf_x F(x) \le \Delta,$$

$$\|\nabla F(x) - \nabla F(y)\| \le L\|x - y\|, \forall x, y \in \mathbb{R}^d \big\} \tag{2}$$

where $\Delta \ge 0$ and $L > 0$ are fixed parameters. The condition $F(0) - \inf_x F(x) \le \Delta$ on $F(0)$ can be generalized to any initial value $F(x_0)$. However, for *zero-respecting* algorithms (to be defined in Section 2.2), we have $x_0 = 0$. In particular, we consider the case where the objective $F$ is smooth and has bounded initial gap to the optimum.

### 2.2 Algorithm class

Our algorithm class is inspired by [5]. We consider the following first-order algorithms such that:

- the algorithm can access an unknown $F \in \mathcal{F}(\Delta, L)$ by a stochastic first-order oracle $O$;
- the oracle $O$ returns a sequence of samples $z := \{s_i\}_{i=1}^{B}$ ($B$ can be time-dependent) generated by a Markov chain and a mapping

$$O_F(x, \{s_i\}_{i=1}^{B}) := \{g(x; s_i))\}_{i=1}^{B}$$

  where $g(x; s) := \nabla f(x; s)$ is the stochastic gradient.

- at iteration $t$, the algorithm queries a batch of $M$ points[1]

$$x_t := (x_{t,1}, x_{t,2}, \ldots, x_{t,M});$$

- for each batch query $x_t$, $O$ responses with

$$O_F(x_t, z_t) := (O_F(x_{t,1}, z_{t,1}), \ldots, O_F(x_{t,M}, z_{t,M})),$$

where $z_{t,i}$ is the sequence of sample drawn for $x_{t,i}$ and $z_t := \bigcup_{i=1}^M z_{t,i}$.

Then algorithm $\mathcal{A}$ consists of a sequence of measurable mappings $\{\mathcal{A}_t\}_{t=0}^\infty$ to generate a sequence of iterates $\{x_t\}_{t=0}^\infty$ satisfying the following conditions:

- the $(t+1)$-th iterate is the output of $\mathcal{A}_t$ when taking all previous oracle responses as input, i.e.,

$$x_{t+1}^{\mathcal{A}[O_F]} = \mathcal{A}_t \left( O_F(x_0^{\mathcal{A}[O_F]}, z_0), \ldots, O_F(x_t^{\mathcal{A}[O_F]}, z_t) \right);$$

- Algorithm $\mathcal{A}$ is *zero-respecting*, i.e., for any $O$ and samples $z_0, z_1, \ldots$ with any $M$, it satisfies for any $t \geq -1$ and any $m \in [M]$

$$\mathrm{support}(x_{t+1,m}^{\mathcal{A}[O_F]}) \subseteq \bigcup_{k \leq t, m' \in [M]} \mathrm{support}(g_{k,m'}), \tag{3}$$

where $g_{k,m'}$ is the stochastic gradient for $x_{k,m'}^{\mathcal{A}[O_F]}$ and $\mathrm{support}(x)$ represents non-zero coordinates of $x$.

We denote $\mathbf{A}_{zr}(M)$ the class of all zero-respecting algorithms. It is worth noting that for any $\mathcal{A} \in \mathbf{A}_{zr}(M)$, $x_{0,1}^{\mathcal{A}[O_F]} = 0$ by definition.

We note that the above-mentioned algorithm class is general, which captures many existing first-order algorithms. For example, the vanilla MC-SGD [14]

$$x_{t+1} = x_t - \eta_t g(x_t; s_t)$$

corresponds to $M = 1, B = 1, \forall t \geq 0$. For Randomized ExtraGradient [6], which maintains the update by

$$x_{t+1/2} = x_t - \eta_t g(x_t; s_{T_t+1})$$
$$x_{t+1} = x_t - \eta_t u_t$$

where by generating $J_t \sim \mathrm{Geom}(1/2)$

$$u_t = u_t^0 + \begin{cases} 2^{J_t}(u_t^{J_t} - u_t^{J_t-1}) & \text{if } 2^{J_t} \leq K \\ 0 & \text{otherwise} \end{cases}$$

with

$$u_t^j := 2^{-j} \sum_{i=1}^{2^j} g(x_{t+1/2}; s_{T_t+i+1}), \quad T_{t+1} = T_t + 1 + 2^{J_t} \text{ for } T_0 = 0,$$

one can clearly see that it fits in the case of $M = 2$ and $B = 2^{J_t}$.

## 2.3 Markov-Chain Classes

In this section, we are interested in any sampling schemes characterized by finite-state Markov chains, i.e., $|\mathcal{S}| < \infty$. Before formally defining the classes of Markov chains, we present the definitions of hitting and mixing times which are two critical quantities characterizing the structure of a chain.

**Definition 2.1** (Hitting time). For any state $w \in \mathcal{S}$, define

$$\tau_w := \inf\{t \geq 1 \mid s_t = w\}$$

as the time Markov chain firstly reaches state $w$. The hitting time $\tau_{hit}$ is defined by

$$\tau_{hit} := \max_{v,w \in \mathcal{S} \times \mathcal{S}} \mathbb{E}[\tau_w \mid s_0 = v].$$

---

[1]The batched queries represent multiple variables maintained by the algorithm to update iteratively at every iteration.

Intuitively, the hitting time measures the maximal number of steps for which any pair of states take to transit between each other.

**Definition 2.2** (Mixing time). Consider Markov chain $P$. For any $\epsilon > 0$, define

$$t_{mix}(\epsilon) := \inf\{\tau \geq 0 \mid d_{TV}(P^\tau, \pi) \leq \epsilon\}$$

where $d_{TV}(P, Q)$ represents the total variation distance between probability measures $P$ and $Q$. We call $\tau_{mix}$ the mixing time of $P$, where $\tau_{mix} := t_{mix}(1/4)$.

The mixing time measures the convergence speed of a Markov chain to its stationary distribution. Then we consider the class of Markov chains for which the stationary distribution $\pi$ exists and the hitting time $\tau_{hit}$ is upper bounded by parameter $\tau \geq 1$ (scaled by some numerical constant). We denote the chain by $P$. Specifically,

$$\mathcal{M}(\tau) := \left\{ P \mid \tau_{hit} \leq c_0 \tau, \lim_{t \to \infty} \mu P^t = \pi, \forall \mu \right\} \tag{4}$$

where $c_0 > 0$ is some numerical constant; $\tau_{hit}$ is defined in Definition 2.1; $\mu P^t$ represents the distribution of the chain after $t$-step transitions starting from the initial distribution $\mu$.

## 2.4 Oracle Classes

Recalling that the oracle $O$ returns a sequence of stochastic gradient evaluated at each query, we place the following assumption.

**Assumption 2.3.** There exists some $0 < \sigma^2 < \infty$, such that $\mathbb{E}_{s \sim \pi} \|g(x; s) - \nabla F(x)\|^2 \leq \sigma^2$ and $\mathbb{E}_{s \sim \pi}[g(x; s)] = \nabla F(x)$ for any $x$.

Basically, Assumption 2.3 only limits the asymptotic behaviors of the stochastic gradient: 1) gradient estimate, i.e., $\mathbb{E}_{s \sim \pi}[g(x; s)] = \nabla F(x)$; and 2) bounded variance, i.e., $\mathbb{E}_{s \sim \pi} \|g(x; s) - \nabla F(x)\|^2 \leq \sigma^2$. Note that this assumption does not place any restriction when the chain has not yet converged to its stationary distribution. Moreover, it becomes aligned with the bounded variance assumption of stochastic first-order methods under i.i.d. sampling by further forcing independence across samples [18, 4].

Then, we are interested in two oracle classes. The first natural class, denoted by $\mathbb{O}(\sigma^2, \tau)$, is that the stochastic gradient $g$ is sampled from a chain contained in $\mathcal{M}(\tau)$ by (4) and such that Assumption 2.3 is satisfied. This oracle class is considered in SGD-based analysis in literature [6].

The second class we consider is inspired by applications of variance reduction in the i.i.d. setting [16, 5], where besides bounded variance of stochastic gradients, an stronger requirement is placed. In particular, we assume that the stochastic gradient $g$ satisfies the mean-squared smoothness:

**Assumption 2.4.** There exists $\bar{L} > 0$ such that $\mathbb{E}_{s \sim \pi} \|g(x; s) - g(y; s)\|^2 \leq \bar{L}^2 \|x - y\|^2, \forall x, y \in \mathbb{R}^d$.

It is straightforward to observe that mean-squared smoothness implies smoothness of $F$ by observing $\|\mathbb{E}_{s \sim \pi}[g(x; s) - g(y; s)]\|^2 \leq \mathbb{E}_{s \sim \pi} \|g(x; s) - g(y; s)\|^2 \leq \bar{L}^2 \|x - y\|^2$. Then, in the second oracle class, denoted by $\mathbb{O}(\sigma^2, \tau, \bar{L}^2)$, we force the stochastic gradient $g$ is sampled from a chain in $\mathcal{M}(\tau)$ and such that both Assumptions 2.3 and 2.4 are satisfied.

## 2.5 Sample Complexity Measures

Our results of lower bounds are established in terms of the sample complexity for finding an $\epsilon$-approximate critical solution of $F$. Define $\mathcal{P}$ as the collection of all well-defined probability measures supported on $\mathcal{S}$. Let $S_t(\mathcal{A}) = \bigcup_{s \leq t} z_t$ be the collection of all samples utilized til time $t$ by algorithm $\mathcal{A}$.

Concretely, the sample complexity measure for the smooth setting is defined by

$$\begin{aligned} &N^\epsilon(M, \Delta, L, \sigma^2, \tau) \\ &:= \sup_{\pi \in \mathcal{P}} \sup_{O \in \mathbb{O}(\sigma^2, \tau)} \sup_{F \in \mathcal{F}(\Delta, L)} \inf_{\mathcal{A} \in \mathbf{A}_{zr}(M)} \\ &\quad \inf \left\{ |S_T(\mathcal{A})| \geq 1 \mid \mathbb{E}\|\nabla F(x_{T,1}^{\mathcal{A}[O_F]})\| \leq \epsilon \right\}. \end{aligned} \tag{5}$$

Similarly, for the mean-squared smooth setting, the sample complexity measure is given by

$$N^\epsilon(M, \Delta, \bar{L}^2, \sigma^2, \tau)$$

$$:= \sup_{\pi \in \mathcal{P}} \sup_{O \in \mathbb{O}(\sigma^2, \tau, \bar{L}^2)} \sup_{F \in \mathcal{F}(\Delta, L)} \inf_{\mathcal{A} \in \mathbf{A}_{zr}(M)}$$

$$\inf \left\{ |S_T(\mathcal{A})| \geq 1 \mid \mathbb{E} \|\nabla F(x_{T,1}^{\mathcal{A}[O_F]})\| \leq \epsilon \right\}. \tag{6}$$

When $N^\epsilon(M, \Delta, L, \sigma^2, \tau)$ is lower bounded by $N_T$, i.e., $N^\epsilon(M, \Delta, L, \sigma^2, \tau) \geq N_T$ with $N_T$ denoting all collected samples up to time $T$, it indicates that there exists some stationary Markov sampling process $P$ with bounded hitting time and an oracle $O \in \mathbb{O}(\sigma^2, \tau)$ such that for any $\mathcal{A} \in \mathbf{A}_{zr}(M)$ there exists $F \in \mathcal{F}(\Delta, L)$ for which $\mathbb{E} \|\nabla F(x_{T,1}^{\mathcal{A}[O_F]})\| > \epsilon$, where the expectation is taken over randomness in $\mathcal{A}$ and $O$. In other words, at least $N_T$ number of samples must be required to (possibly) achieve an $\epsilon$-approximate critical solution for any first-order algorithm (and similarly for the mean-squared smooth setting).

## 3 Algorithm-independent Lower Bounds

In this section, we show our main results on the lower bounds of sample complexity for stochastic non-convex optimization under Markov sampling. The result is algorithm-independent, implying that all first-order methods that are zero-respecting take at least such samples to reach an $\epsilon$-approximate critical point of the non-convex objective function.

We show the following sample complexity lower bounds for any finite Markov sampling processes under both smooth and mean-squared smooth settings. Please refer to Appendices B and C for detailed proof.

**Theorem 3.1.** *For the smooth setting, there exist numerical constants $c_1, c_2 > 0$ such that for any $M, L, \Delta, \sigma, \tau > 0$,*

$$N^\epsilon(M, \Delta, L, \sigma^2, \tau) = \Omega \left( \frac{\tau L \Delta}{\epsilon^2} + \frac{\tau \sigma^2}{\epsilon^4} \min \left\{ c_1 \sigma^2, c_2 L \Delta \right\} \right).$$

*For the mean-squared smooth setting, we have*

$$N^\epsilon(M, \Delta, \bar{L}^2, \sigma^2, \tau) = \Omega \left( \frac{\tau \bar{L} \Delta}{\epsilon} + \frac{\tau \bar{L} \Delta \sigma^2}{\epsilon^3} \right).$$

*Remark* 3.2. Note that the extreme case $\tau = 1$ corresponds to the i.i.d. sampling case. To see this, recalling the definition of hitting time $\tau = 1$ indicates exactly one step is taken transiting from one state to any other, which then implies the samples are drawn exactly from the stationary distribution $\pi$ and there is no time dependence across samples drawn at different time steps, hence reducing to i.i.d. case. Thus, when $\sigma^2 \succeq L\Delta$ our lower bound results are aligned with those provided in [5]. Moreover, noting that for the smooth setting the lower bound reduces to $\Omega \left( \frac{\tau L \Delta}{\epsilon^2} + \frac{\tau \sigma^2 L \Delta}{\epsilon^4} \right)$ if $\sigma^2 \succsim L\Delta$, it nearly matches the best-known upper bound in literature [6], which is $\tilde{\mathcal{O}} \left( \frac{\tau_{mix} L \Delta}{\epsilon^2} + \frac{\tau_{mix} \sigma^2 L \Delta}{\epsilon^4} \right)^2$

## 4 Min-max Optimality for Mean-squared Smooth Functions

In this section, we propose **Ma**rkov-**C**hain **SPIDER** (MaC-SPIDER) ,which is a variant of SPIDER [16] under Markov sampling, manifesting the (near) min-max optimality of sample complexity. In particular, we show that the sample complexity of MaC-SPIDER is $\mathcal{O}(\tau_{mix} \pi_{min}^{-1/2} \epsilon^{-3})$ (with $\pi_{min} = \min_s \pi(s)$), which is comparable to our proposed lower bound $\Omega(\tau \epsilon^{-3})$ up to some constant gap (independent of $\epsilon$) between $\pi_{min}^{-1/2} \tau_{mix}$ and the hitting time $\tau$. Proof for this section is deferred to Appendix D.

We present MaC-SPIDER in Algorithm 1, which exhibits effective variance reduction ability, hence improving sample complexity. Algorithmically, it is similar to SPIDER [16], while now the sampled

---

[2]Note that this upper bound is achieved under a stronger assumption than Assumption 2.3, i.e., in [6] it is assumed that $\|g(x; s) - \nabla F(x)\|^2 \leq \sigma^2, \forall x, s$. But we conjecture this bound also holds under Assumption 2.3.

---

**Algorithm 1** Markov-Chain **SPIDER** (MaC-SPIDER)

---

1: **Input:** initial point $x_0$, $N_0 = 0$, batch size $M_1$ and $M_2$, integers $T$ and $r$, stepsizes $\{\eta_t\}_{t=0}^{T-1}$
2: **for** $t = 0, 1, \ldots, T - 1$ **do**
3:    **if** $t \bmod r = 0$ **then**
4:       Draw $M_1$ samples and compute $v_t = \frac{1}{M_1} \sum_{i=1}^{M_1} g(x_t; s_{N_t+i})$.
5:       $N_{t+1} = N_t + M_1$.
6:    **else**
7:       Draw $M_2$ samples and compute $v_t = v_{t-1} + \frac{1}{M_2} \sum_{i=1}^{M_2} (g(x_t; s_{N_t+i-1}) - g(x_{t-1}; s_{N_t+i}))$.
8:       $N_{t+1} = N_t + M_2$.
9:    **end if**
10:   Set the learning rate $\eta_t = \min\left\{ \frac{1}{8L}, \frac{\epsilon}{4L\|v_t\|} \right\}$.
11:   $x_{t+1} = x_t - \eta_t v_t$.
12: **end for**
13: **Output:** $\tilde{x}_T$ sampled uniformly from $\{x_t\}_{t=0}^{T-1}$.

---

gradients are drawn from some unknown Markov chain rather than identically and independently. One crucial characteristic of MaC-SPIDER is that only one single trajectory generated by the chain is used for implementation. In other words, one needs not to care about when or where to restart the chain once the sampling process begins, but just keeps in mind how many samples should be drawn every iteration. This also enables its broader application in practice. For example, in literature [30, 26] multiple trajectories must be simulated before variance-reduced techniques to be applied, which costs extra resources and waiting time since every trajectory requires restart and then wait until enough collected samples. More problematically, multiple trajectories may even become unavailable in practical scenarios, when the chain is uncontrollable to simulate. MaC-SPIDER only uses a single trajectory with no further limitation and knowledge on the chain, which in this sense distinguishes its simplicity and applicability for practical implementation.

In particular, $v_t$ maintained in Algorithm 1 serves as an estimate of $\nabla F(x_t)$ that incorporates the history of sampled gradients for a better estimation with fewer samples to control the variance. This is intuitive and understandable when samples are i.i.d., as each sample is an independent and unbiased estimate of the true gradient. However, such unbiasedness no longer holds in our case, since Markovian samples are essentially biased and time-dependent, which makes it unknown whether or not variance can still be reduced effectively. Fortunately, by carefully analyzing the coupled correlation across different samples, we provide a positive answer to the controllability of the variance of $v_t$, formally stated by Proposition 4.3.

We first present the following useful lemma, which generally provide the basic guidance on the variance control of Markov sampling.

**Lemma 4.1.** *For a Markov chain $\{s_t\}_{t=0}^{\infty}$ with state space $\mathcal{S}$ and stationary distribution $\pi$, consider any function $h : \mathcal{S} \rightarrow \mathbb{R}^d$ and define $h_\pi := \mathbb{E}_{s\sim\pi}[h(s)]$. Denoting $\mathbb{E}_t(\cdot)$ as the expectation conditioning on filtration $\mathcal{F}_t$ and $\tau_{mix}$ as the mixing time, the following holds:*

*(1). If $\mathbb{E}_{s\sim\pi}\|h(s) - h_\pi\|^2 \leq \sigma^2$, given any $s_t \in \mathcal{S}$,*

$$\mathbb{E}_t \left\| \frac{1}{M} \sum_{i=1}^{M} h(s_{t+i}) - h_\pi \right\|^2 \leq \frac{7\tau_{mix}\sigma^2}{\sqrt{\pi_{min}}M} + \frac{78\tau_{mix}^2\sigma^2}{\pi_{min}M^2}. \tag{7}$$

*(2). If $\mathbb{E}_{s\sim\pi}\|h(s)\|^2 \leq B^2$, then for any $s_t$*

$$\left\| \mathbb{E}_t \left( \frac{1}{M} \sum_{i=1}^{M} h(s_{t+i}) - h_\pi \right) \right\| \leq \frac{6\tau_{mix}B}{\sqrt{\pi_{min}}M}, \tag{8}$$

$$\mathbb{E}_t \left\| \frac{1}{M} \sum_{i=1}^{M} h(s_{t+i}) - h_\pi \right\|^2 \leq \frac{14\tau_{mix}B^2}{\sqrt{\pi_{min}}M} + \frac{228\tau_{mix}^2B^2}{\pi_{min}M^2}. \tag{9}$$

*Remark* 4.2. Note that similar variance bounds as Part (1) are provided in [6, 11]. However, either of their analysis requires bounded gradient assumption, i.e., $\|\nabla F(x)\| \leq G < \infty, \forall x$ or bounded

noise assumption, i.e., $\|g(x; s) - \nabla F(x)\|^2 \le \sigma^2 < \infty, \forall x, s$, while ours do not depend on such restrictive assumptions.

Then leveraging Lemma 4.1 gives the following result, which indicates that the variance of $v_t$ can be well controlled to an arbitrary level.

**Proposition 4.3.** *Considering Algorithm 1 and supposing Assumptions 2.3, 2.4 hold, by setting*

$$M_1 = \frac{112\tau_{mix}}{\sqrt{\pi_{min}}\epsilon^2} \max\{\sigma, \sigma^2\}, \quad M_2 = \frac{16\tau_{mix}}{\sqrt{\pi_{min}}\epsilon}, \quad r = \frac{1}{\epsilon}$$

*we have for any $\epsilon \le 1$*

$$\mathbb{E}\|v_t - \nabla F(x_t)\|^2 \le \epsilon^2, \quad \forall t \ge 0.$$

It is worth noting that Proposition 4.3 implies that to control the variance to a level of $\mathcal{O}(\epsilon^2)$, the average number of samples per iteration that Algorithm 1 uses is $\mathcal{O}(\epsilon^{-1})$, while simply using batched samples requires $\mathcal{O}(\epsilon^{-2})$ samples [11, 6]. Thus MaC-SPIDER successfully reduces the variance with fewer samples. This is the main reason why the sample complexity can be improved to $\mathcal{O}(\epsilon^{-3})$. It is formally summarized by the following theorem.

**Theorem 4.4.** *For Algorithm 1 with any $\epsilon \le 1$ assuming Assumptions 2.3, 2.4 hold, setting*

$$M_1 = \frac{112\tau_{mix}}{\sqrt{\pi_{min}}\epsilon^2} \max\{\sigma, \sigma^2\}, \quad M_2 = \frac{16\tau_{mix}}{\sqrt{\pi_{min}}\epsilon}, \quad r = \frac{1}{\epsilon}, \quad T = \frac{16\bar{L}\Delta}{\epsilon^2}$$

*where $\Delta = F(x_0) - \min_x F(x)$, we guarantee*

$$\mathbb{E}\|\nabla F(\tilde{x}_T)\| \le 7\epsilon.$$

*Moreover, the total number of samples is $\mathcal{O}(\tau_{mix}\pi_{min}^{-1/2}\epsilon^{-3})$.*

*Remark* 4.5. Combining with the lower bound in Theorem 3.1, we note that MaC-SPIDER achieves almost the same order of $\epsilon^{-3}$ up to some gap (independent of $\epsilon$) between $\tau$ and $\pi_{min}^{-1/2}\tau_{mix}$. This shows its near min-max optimality.

# 5 Proof Idea of the Lower Bounds

In this section we present the proof idea of how we obtain the sample complexity lower bounds for Markov sampling. We first clarify the proof sketch by focusing on the case where $B = 1$, i.e., only one sample is drawn from the underlying Markov chain by the algorithm. Then, we generalize it to the case of $B \ge 1$ which can also be time-dependent. Full proofs are presented in Appendices B and C.

The core technique inspired by [5] is to construct a "hard" function $F$ with $f(\cdot; s)$ supported on each state of a Markov chain lying in the required class such that the gradient norm, $\|\nabla F(x)\|$, is small only if each coordinate of $x$ has a large enough absolute value. We use the progress function to mathematically evaluate the largest coordinate whose absolute value is larger than some nonnegative scalar $\alpha$, i.e.,

$$\text{prog}_\alpha(x) := \max\{k \ge 1 \mid |[x]_k| > \alpha\}$$

where $[x]_k$ represents the $k$-th coordinate of $x$. We set $\text{prog}_\alpha(x) = 0$ if $|[x]_k| \le \alpha, \forall k \in [d]$. Then the task of finding an $\epsilon$-approximate critical solution is equivalently transformed to finding a solution $x$ whose coordinate progress is high. Formally it is stated by the following lemma.

**Lemma 5.1.** *There exists some $F^* \in \mathcal{F}(\mathcal{O}(\Delta\epsilon^2 d), L)$ such that $\|\nabla F^*(x)\| > \epsilon, \forall \epsilon > 0$ if $\text{prog}_0(x) < d$.*

Indicated by Lemma 5.1 ensuring $\|\nabla F^*(x)\| \le \epsilon$ requires all coordinates of $x$ to be nonzero. Then for the smooth setting, we construct a chain with its hitting time upper bounded by $\tau$ and at least $\Omega(\tau/q)$ (with $q \in (0, 1)$) iterations are needed to make one increase in $\text{prog}_0(x)$. It is stated by the following lemma.

**Lemma 5.2.** *For any $q \in (0, 1)$ and any zero-respecting algorithm $\mathcal{A} \in \mathbf{A}_{zr}$, there exist a Markov chain contained in $\mathcal{M}(\tau)$ and some $F^* \in \mathcal{F}(\mathcal{O}(\Delta d\epsilon^2), L)$ with $g^*(x; s)$ satisfying $\nabla F^*(x) = \mathbb{E}_{s \sim \pi}[g^*(x; s)]$ and $\mathbb{E}_{s \sim \pi}\|g^*(x; s) - \nabla F^*(x)\|^2 \le \mathcal{O}(\sigma^2\epsilon^2/q)$ such that for any $0 < \delta < 1$, with probability at least $1 - \delta$*

$$\max_{m \in [M]} \max_{s \le t} \text{prog}_0(x_{s,m}^{\mathcal{A}[O_F]}) < d, \quad \forall t \le \frac{\tau(d - \log \delta^{-1})}{4q}.$$

In fact the constructive function $F^*$ in Lemma 5.2 coincides with the one in Lemma 5.1, which then implies that at least $\Omega(\tau d/q)$ iterations are needed to guarantee an $\epsilon$-approximate critical solution output by any algorithm and hence $\Omega(\tau d/q)$ samples (due to $B = 1$). Finally setting $d = \Omega(\epsilon^{-2})$ and $q = \mathcal{O}(\epsilon^2)$ concludes the lower bound $\Omega(\tau\epsilon^{-4})$ shown by the smooth setting of Theorem 3.1.

For the mean-squared smooth setting, we have the following result.

**Lemma 5.3.** *For any $q \in (0, 1)$ and any zero-respecting algorithm $\mathcal{A} \in \mathbf{A}_{zr}$, there exist a Markov chain contained in $\mathcal{M}(\tau)$ and some $\bar{F}^* \in \mathcal{F}(\mathcal{O}(\Delta d\epsilon), L)$ with $\bar{g}^*(x; s)$ satisfying $\nabla \bar{F}^*(x) = \mathbb{E}_{s\sim\pi}[\bar{g}^*(x; s)]$ and $\mathbb{E}_{s\sim\pi}\|\bar{g}^*(x; s) - \nabla \bar{F}^*(x)\|^2 \leq \mathcal{O}(\sigma^2\epsilon^2/q)$ such that for any $0 < \delta < 1$, with probability at least $1 - \delta$*

$$\max_{m\in[M]} \max_{s\leq t} prog_0(x_{s,m}^{\mathcal{A}[O_F]}) < d, \ \ \forall t \leq \frac{\tau(d - \log\delta^{-1})}{4q}.$$

Thus, taking $d = \mathcal{O}(\epsilon^{-1})$ and $q = \mathcal{O}(\epsilon^2)$ recovers the bound for the mean-squared smooth setting in Theorem 3.1.

Note that the above proof derivations are established on the precondition when $B = 1$ by which we are able to directly obtain the sample complexity bounds through the iteration complexity analysis, since the iteration complexity is the same as the sample complexity. To generalize our results to $B \geq 1$, we present the following result.

**Lemma 5.4.** *There exist a Markov chain in* (4) *and some functions $F^*, g^*$ $(\bar{F}^*, \bar{g}^*)$ satisfying corresponding conditions in Lemma 5.2 (or Lemma 5.3) such that for any zero-respecting algorithm $\mathcal{A}_{zr}$ with $B \geq 1$, there is a zero-respecting algorithm $\mathcal{A}_{zr}^*$ with $B = 1$ for which the following holds: for any $t \geq 0$ if $\max_{m\in[M]} \max_{s\leq t} prog_0(x_{s,m}^{\mathcal{A}^*[O_F]}) \leq k$, then $\max_{m\in[M]} \max_{s\leq t} prog_0(x_{s,m}^{\mathcal{A}[O_F]}) \leq k, \forall 0 \leq k \leq d$.*

The above lemma indicates that we can always find an algorithm that only draws one sample per iteration to achieve no worse progress in its update than other algorithms that access multiple samples per iteration. In other words, combining with Lemmas 5.1 and 5.2 yields that accessing multiple samples every iteration has no benefit on improving the sample complexity for the algorithm, which hence implies the lower bounds that holds for $B = 1$ also holds for $B \geq 1$.

## 6  Limitation

The paper only considers the case of finite-state Markov chains, while in practice it is highly possible that the underlying Markovian sampling process admits infinitely many states. Besides, we only consider stationary Markov chains in the paper, which limits the generalization to the non-stationary setting. Therefore, extension to non-stationary Markov chains is an interesting direction and may need extra efforts.

## 7  Conclusion

In this paper, we study the first-order non-convex stochastic optimization problems. Unlike the conventional i.i.d. sampling, we focus on the case where data samples and stochastic gradient estimates are generated by an unknown Markov chain, which introduces additional data correlation and hence non-trivial analysis difficulties. Due to the lack of sample complexity lower bound results and the gap to the best-known upper bound under the Markovian setting, we provide an improved complexity lower bound with the order of $\epsilon^{-4}$ for general smooth functions, which nearly matches the best-known upper bound. We also consider the mean-squared smooth setting, which exhibits broad applications in variance reduction literature. We prove that for the mean-squared smooth setting, the sample complexity lower bound is the order of $\epsilon^{-3}$ for Markovian samples. Finally, we propose a new algorithm MaC-SPIDER, which to the best of our knowledge is the first variance-reduced method under a single Markovian trajectory, such that its sample complexity upper bound nearly matches our proposed lower bound, implying its near min-max optimality and the tightness of the lower bound.

## Acknowledgements

This work is supported by NSF awards ECCS-2030251, 2216926, CMMI-2024774 and AST-2421845.

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

# A    Constructions of the Markov Chain and "Hard" Function $F$

## A.1    Construction of Markov chains

In this section, we construct a Markov chain that is used for the lower bound proofs. The idea is to construct a chain such that 1) there exist two states between which at least $\tau$ steps must be take to transit; 2) the hitting time of the constructed chain is upper bounded by $\tau$. Without loss of generality we assume $\tau$ is even.

In particular, we consider a directed cyclic-like chain with self-loops. Denote $s = i$ be the $i$-th state of the Markov chain for $i \in \{0, 1, \ldots, 2\tau' + 1\}$ with $\tau' = \tau/2$. Then for any $q \in (0, 1/2)$ the transition of the chain is defined as follows:

- $P(s = 2|s = 0) = P(s = 0|s = 0) = P(s = 2|s = 1) = P(s = 1|s = 1) = 1/2$;
- $P(s = \tau' + 3|s = \tau' + 1) = P(s = \tau' + 1|s = \tau' + 1) = P(s = \tau' + 3|s = \tau' + 2) = P(s = \tau' + 2|s = \tau' + 2) = 1/2$;
- $P(s = \tau' + 1|s = \tau') = q, P(s = \tau' + 2|s = \tau') = 1/2 - q, P(s = \tau'|s = \tau') = 1/2$;
- $P(s = 0|s = 2\tau' + 1) = q, P(s = 1|s = 2\tau' + 1) = 1/2 - q, P(s = 2\tau' + 1|s = 2\tau' + 1) = 1/2$;
- $P(s = i + 1|s = i) = P(s = i|s = i) = 1/2, \forall i \notin \{0, 1, \tau', \tau' + 1, \tau' + 2, 2\tau' + 1\}$.

Then letting $v_1^* = 0, v_2^* = 1, w_1^* = \tau' + 1, w_2^* = \tau' + 2$, it is straightforward that the above constructed Markov chain guarantees that transitioning between $v_1^*$ and $w_1^*$ takes at least $\tau' = \tau/2$ steps. Moreover, the hitting time of the chain is $\mathcal{O}(\tau)$ by noting the hitting time of directed cyclic chain with self-loops and $n$ states is $\mathcal{O}(n)$. We denote this chain by $P^*$, and hence $P^* \in \mathcal{M}(\tau)$.

## A.2    Construction of function $F$

Now we construct a "hard" function that is difficult for any first-order algorithm to search for the critical point. Specifically we consider the following two functions

$$h_1(x) = -\psi(1)\phi([x]_1) + \sum_{i=1}^{\lfloor d/2 \rfloor - 1} (\psi(-[x]_{2i})\phi(-[x]_{2i+1}) - \psi([x]_{2i})\phi([x]_{2i+1})) \qquad (10)$$

$$h_2(x) = \sum_{i=1}^{\lfloor d/2 \rfloor} (\psi(-[x]_{2i-1})\phi(-[x]_{2i}) - \psi([x]_{2i-1})\phi([x]_{2i})) \qquad (11)$$

where

$$\psi(u) = \begin{cases} 0 & , \quad u \le \frac{1}{2} \\ \exp\left(1 - \frac{1}{(2u-1)^2}\right) & , \quad u > \frac{1}{2} \end{cases}$$

and

$$\phi(u) = \sqrt{e} \int_{-\infty}^{u} e^{-\frac{t^2}{2}} dt$$

with $u \in \mathbb{R}$.

We denote $\pi_s$ as the corresponding probability of state $s$ of the stationary distribution $\pi$. Then, given the Markov chain $P^*$ constructed in Appendix A.1, we know that at least $\frac{1}{2}\tau$ steps are required to take transiting from $v_1^*$ to $w_1^*$ and vice versa. Then, we construct function $F$ such that $F(x) = \pi_{v^*} h_1(x) + \pi_{w^*} h_2(x)$, where we denote $v^* = \{v_1^*, v_2^*\}$, $w^* = \{w_1^*, w_2^*\}$ and $\pi_{v^*} = \pi_{v_1^*} + \pi_{v_2^*}$, $\pi_{w^*} = \pi_{w_1^*} + \pi_{w_2^*}$. For any $x$ and $i \ge 0$ define $x_{\le i} := ([x]_1, \ldots, [x]_i, 0, \ldots, 0)$ as the truncated version by only keeping the first $i$ coordinates. We also set $x_{\le 0} = x$. Then we have the following properties of $F$.

**Lemma A.1.** *Let* $F(x) = \pi_{v^*} h_1(x) + \pi_{w^*} h_2(x)$ *for* $h_1, h_2$ *defined by* (10),(11). *Then we have the following:*

*(1).* $F(0) - \inf_x F(x) \le \Delta_0 d$ *for some constant* $\Delta_0 > 0$.

*(2).* $\|\nabla h_i(x)\|_\infty \le 23$ *and* $\|\nabla h_i(x)\| \le 23\sqrt{d}, i = 1, 2$.

*(3). $F(x)$ is $l_1$-smooth for some constant $l_1 > 0$.*

*(4). If $\text{prog}_1(x) < d$, $\|\nabla F(x)\| \geq 1$.*

*(5). $[\nabla h_i(x)]_{\leq \text{prog}_{\frac{1}{2}}(x)} = [\nabla h_i(x_{\leq \text{prog}_{\frac{1}{2}}(x)})]_{\leq \text{prog}_{\frac{1}{2}}(x)}$, $i = 1, 2$.*

*(6). If $\text{prog}_0(x)$ is odd, $\text{prog}_0(\nabla h_1(x)) \leq \text{prog}_{\frac{1}{2}}(x)$, $\text{prog}_0(\nabla h_2(x)) \leq \text{prog}_{\frac{1}{2}}(x) + 1$. If $\text{prog}_0(x)$ is even, $\text{prog}_0(\nabla h_1(x)) \leq \text{prog}_{\frac{1}{2}}(x) + 1$, $\text{prog}_0(\nabla h_2(x)) \leq \text{prog}_{\frac{1}{2}}(x)$.*

*(7). If $\text{prog}_{\frac{1}{2}}(x)$ is odd, $\nabla h_1(x) = \nabla h_1(x_{\leq \text{prog}_{\frac{1}{2}}(x)})$, $\nabla h_2(x) = \nabla h_2(x_{\leq 1 + \text{prog}_{\frac{1}{2}}(x)})$. If $\text{prog}_{\frac{1}{2}}(x)$ is even, $\nabla h_1(x) = \nabla h_1(x_{\leq 1 + \text{prog}_{\frac{1}{2}}(x)})$, $\nabla h_2(x) = \nabla h_2(x_{\leq \text{prog}_{\frac{1}{2}}(x)})$.*

*Proof.* For Part (1), observing that $F(0) < 0$ and noting that $0 \leq \psi(u) \leq e$, $0 \leq \phi(u) \leq \sqrt{2\pi e}$,

$$F(x) \geq -\psi(1)\phi([x]_1) - \sum_{i=2}^{d} \psi([x]_{i-1})\phi([x]_i) \geq -de\sqrt{2\pi e} = -d\Delta_0$$

with $\Delta_0 = e\sqrt{2\pi e}$, which completes its proof.

For Part (2), noting that $0 \leq \psi'(u) \leq \sqrt{54e^{-1}}$ and $0 \leq \phi'(u) \leq \sqrt{e}$, combining with the fact that for each $i = 1, 2$

$$\frac{\partial h_i}{\partial x_j}(x) \geq \psi(-[x]_{j-1})\phi'(-[x]_j) - \psi([x]_{j-1})\phi'([x]_j) - \psi'(-[x]_j)\phi(-[x]_{j+1}) - \psi'([x]_j)\phi([x]_{j+1})$$

yields

$$\left|\frac{\partial h_i}{\partial x_j}(x)\right| \leq e\sqrt{e} + \sqrt{54e^{-1}}\sqrt{2\pi e} \leq 23$$

implying $\|\nabla h_i(x)\| \leq 23$ and $\|\nabla h_i(x)\| \leq 23\sqrt{d}$, $\forall i = 1, 2$.

Parts (3) and (4) follow directly from [10]. Parts (5)-(7) follow from the observation that

$$\nabla h_1(x) = \nabla h_1([x]_1, \ldots, [x]_{2i+1}, 0, \ldots, 0), \text{ if } |x_{2j}| \leq \frac{1}{2}, \ \forall j \geq i + 1$$

$$\nabla h_2(x) = \nabla h_2([x]_1, \ldots, [x]_{2i}, 0, \ldots, 0), \text{ if } |x_{2j-1}| \leq \frac{1}{2}, \ \forall j \geq i + 1.$$

$\square$

# B  Lower Bound for the Smooth Setting

In this section, we show the lower bound of the smooth setting in Theorem 3.1. Based on the contructive $F$ in the last section, we consider the following gradient oracle $g$: for each $i$-th coordinate of $g$

$$\text{if } s \in \{v_1^*, v_2^*\}, \ [g(x; s)]_i = [\nabla h_1(x)]_i \cdot \left(1 + \mathbb{1}\{i > \text{prog}_{\frac{1}{2}}(x)\}\left(\frac{\mathbb{1}_{s=v_1^*}}{q} - 1\right)\right),$$

$$\text{if } s \in \{w_1^*, w_2^*\}, \ [g(x; s)]_i = [\nabla h_2(x)]_i \cdot \left(1 + \mathbb{1}\{i > \text{prog}_{\frac{1}{2}}(x)\}\left(\frac{\mathbb{1}_{s=w_1^*}}{q} - 1\right)\right),$$

$$\text{otherwise, } g(x; s) = 0. \tag{12}$$

Recalling the definition of $P^*$, we note $\mathbb{P}(s = v_1^* \mid s \in \{v_1^*, v_2^*\}) = \mathbb{P}(s = w_1^* \mid s \in \{w_1^*, w_2^*\}) = q \in (0, 1/2)$. Then, we have the following lemma.

**Lemma B.1.** *Considering stochastic gradient $g(x; s)$ constructed as (12), the following statements hold:*

*(1). For $s \in \{v_1^*, v_2^*, w_1^*, w_2^*\}$, with probability at least $1 - q$, $\text{prog}_0(g(x; s)) \leq \text{prog}_{\frac{1}{2}}(x)$ and $g(x; s) = g(x_{\leq \text{prog}_{\frac{1}{2}}(x)}; s)$ for all $x$.*

*(2). For $s \notin \{v_1^*, v_2^*, w_1^*, w_2^*\}$, with probability 1, $prog_0(g(x; s_t)) \le prog_{\frac{1}{2}}(x)$ and $g(x; s_t) = g(x_{\le prog_{\frac{1}{2}}(x)}; s_t)$ for all $x$.*

*(3). For any $s$, with probability 1, $prog_0(g(x; s)) \le 1 + prog_{\frac{1}{2}}(x)$ and $g(x; s) = g(x_{\le 1 + prog_{\frac{1}{2}}(x)}; s)$ for all $x$.*

*(4). $\mathbb{E}_{s \sim \pi}[g(x; s)] = \nabla F(x)$.*

*Proof.* We firstly show Part (3). Note that by (12) and Part (7) of Lemma A.1, for any $x, s$, $[g(x; s_t)]_i = 0, \forall i > 1 + prog_{\frac{1}{2}}(x)$ in the sense that $[\nabla h_1(x)]_i = [\nabla h_2(x)]_i = 0, \forall i > 1 + prog_{\frac{1}{2}}(x)$, which implies $prog_0(g(x; s)) \le 1 + prog_{\frac{1}{2}}(x)$. Moreover, by Part (7) of Lemma A.1, defining $x' := x_{\le 1 + prog_{\frac{1}{2}}(x)}$ gives $\nabla h_1(x) = \nabla h_1(x')$ and $\nabla h_2(x) = \nabla h_2(x')$. Thus, we obtain $g(x; s) = g(x'; s)$ for any $x, s$, implying Part (3).

For Part (1), we note that when $i \ge 1 + prog_{\frac{1}{2}}(x)$ and $s \in \{v_2^*, w_2^*\}$, $g(x; s) = [\nabla h_j(x)]_{\le prog_{\frac{1}{2}}(x)}$ for $j = 1, 2$, which implies $prog_0(g(x; s)) \le prog_{\frac{1}{2}}(x), \forall s \in \{v_2^*, w_2^*\}$. Further, according to (5) of Lemma A.1, we have $g(x; s) = g(x_{\le prog_{\frac{1}{2}}(x)}; s)$ for $s \in \{v_2^*, w_2^*\}$ and all $x$. Since $P(z = 0) = 1 - q$, hence Part (1) is proved.

Part (2) holds trivially in the sense that $g(x; s_t) = 0$ when $s \notin \{v_1^*, v_2^*, w_1^*, w_2^*\}$. Finally, Part (4) holds since $\mathbb{E}[\mathbb{1}_s/q \mid s \in \{v_1^*, v_2^*\}] = \mathbb{E}[\mathbb{1}_s/q \mid s \in \{w_1^*, w_2^*\}] = 1$. □

Also, we show in the following lemma that $g$ has bounded variance.

**Lemma B.2.** *For $F(x) = \pi_{v^*} h_1(x) + \pi_{w^*} h_2(x)$ and $g$ defined as (12), then for any Markov chain with stationary distribution $\pi$, given any $x \in \mathbb{R}^d$,*

$$\mathbb{E}_{s \sim \pi} \|g(x; s) - \nabla F(x)\|^2 \le a_1 d + a_2 \frac{1 - q}{q}$$

*for some constant $a_1, a_2 > 0$.*

*Proof.* By Part (4) of Lemma B.1, we know $\mathbb{E}_{s \sim \pi}[g(x; s)] = \nabla F(x)$.

Denote $i^* = 1 + prog_{\frac{1}{2}}(x)$. For any $s \in \{v_1^*, v_2^*, w_1^*, w_2^*\}$, we have

$g(x; s) - \nabla F(x) = (0, \ldots, 0, [\nabla h_1(x)]_{i^*} (\mathbb{1}_{s = v_1^*}/q - 1), 0, \ldots, 0) + (1 - \tilde{\pi}_{v^*}) \nabla h_1(x) - \tilde{\pi}_{w^*} \nabla h_2(x)$, if $s \in \{v_1^*, v_2^*\}$

$g(x; s) - \nabla F(x) = (0, \ldots, 0, [\nabla h_2(x)]_{i^*} (\mathbb{1}_{s = w_1^*}/q - 1), 0, \ldots, 0) + (1 - \tilde{\pi}_{w^*}) \nabla h_2(x) - \tilde{\pi}_{v^*} \nabla h_1(x)$, if $s \in \{w_1^*, w_2^*\}$.

When $i^* - 1$ is odd, from Part (6) of Lemma A.1 we know that $[\nabla h_1(x)]_{i^*} = 0$. Therefore,

$$\|g(x; s) - \nabla F(x)\|^2 \le 2\|\nabla h_1(x)\|^2 + 2\|\nabla h_2(x)\|^2 \le 4 \cdot 23^2 d, \quad s \in \{v_1^*, v_2^*\}$$

$$\|g(x; s) - \nabla F(x)\|^2 \le 3|[\nabla h_2(x)]_{i^*}|^2 (\mathbb{1}_{s = w_1^*}/q - 1)^2 + 3\|\nabla h_1(x)\|^2 + 3\|\nabla h_2(x)\|^2$$
$$\le 3 \cdot 23^2 (\mathbb{1}_{s = w_1^*}/q - 1)^2 + 6 \cdot 23^2 d, \quad s \in \{w_1^*, w_2^*\}$$

and

$$\|g(x; s_t) - \nabla F(x)\|^2 = \|\nabla F(x)\|^2 \le 4 \cdot 23^2 d, \quad \text{when } s \notin \{v_1^*, v_2^*, w_1^*, w_2^*\}$$

where we use (2) of Lemma A.1. Combining the above three inequalities, it yields that when $i^* - 1$ is odd, for any Markov chain, any $x, t \ge 0$ and any initial distribution of the chain,

$$\mathbb{E}\|g(x; s_t) - \nabla F(x)\|^2 \le a_1 d + a_2 \frac{1 - q}{q}$$

where $a_1 = 6 \cdot 23^2, a_2 = 3 \cdot 23^2$ and we use that $\mathbb{E}[(\mathbb{1}_s/q - 1)^2 \mid s \in \{w_1^*, w_2^*\}] = (1 - q)/q$. The case when $i^* - 1$ is even can be derived similarly. □

Then, we are ready to show Lemmas 5.1 and 5.2. We first focus on the case $B = 1$ and then generalize it to $B \geq 1$.

*Proof of Lemmas 5.1 and 5.2:* Given any $\epsilon > 0$, we consider the following $F^*$

$$F^*(x) := \frac{L\lambda^2}{l_1} F\left(\frac{x}{\lambda}\right), \quad \text{where } \lambda = \frac{2l_1}{L}\epsilon. \tag{13}$$

And we consider the following gradient $g^*$

$$g^*(x; s) := \frac{L\lambda}{l_1} g\left(\frac{x}{\lambda}; s\right)$$

with $g(x; s)$ defined as (12). Since $\nabla F(x) = \mathbb{E}_{s \sim \pi}[g(x; s)]$, $\nabla F^*(x) = \mathbb{E}_{s \sim \pi}[g^*(x; s)]$. We note that

$$\nabla^2 F^*(x) = \frac{L}{l_1} \nabla^2 F\left(\frac{x}{\lambda}\right)$$

which implies that $F^*$ is $L$-smooth by Part (3) of Lemma A.1. Moreover, by Part (1) of Lemma A.1 we obtain that

$$F^*(0) - \inf_x F^*(x) = \frac{4l_1\epsilon^2}{L}(F(0) - \inf_x F(x)) \leq \frac{4l_1\Delta_0\epsilon^2}{L}d.$$

All the above concludes Lemma 5.1.

To see Lemma 5.2, note by Lemma B.2, we have

$$\mathbb{E}_{s \sim \pi}\|g^*(x; s) - \nabla F^*(x)\|^2 \leq 4a_1 d\epsilon^2 + \frac{4a_2(1-q)}{q}\epsilon^2.$$

Then, define

$$B_t := \mathbb{1}\left\{\exists\, x \,:\, \text{prog}_0(g^*(x; s_t)) = 1 + \text{prog}_{\frac{1}{2}}(x)\right\}.$$

Note that under the construction of the Markov chain $P^*$ and $F^*$ and $g^*$, for any zero-respecting algorithm $\mathcal{A}$

$$B_{t+k} = 0, \quad \forall k = 1, \ldots, \frac{1}{2}\tau, \quad \text{conditioning on } B_t = 1.$$

That is to say within every $\frac{1}{2}\tau$ iterations $B_t$ can be 1 at most once. And Part (1) of Lemma B.1 indicates that the probability of $B_t$ being 1 is no greater than $q$. Let $k(t) := \max_{m \in [M]} \max_{l \leq t} \text{prog}_0(x_{l,m}^{\mathcal{A}[O]_{F^*}})$. Then, the above implies that

$$k(t) \leq \sum_{l \leq t} B_l.$$

Also recalling the definition of $P^*$ guarantees for any $t$ in the ideal case the number of possible non-zero $B_l$ can be at most $2t/\tau$ with each being 1 with probability at most $q$, it implies

$$\sum_{l \leq t} B_l \leq \sum_{i=1}^{\lceil 2t/\tau \rceil} z_i$$

where $z_i$ denotes the Bernoulli random variable with succeeding probability being at most $q$. Note that $z_i$s are independent in the sense that conditioning on the chain hits $v^* = \{v_1^*, v_2^*\}$ and will hit $w^* = \{w_1^*, w_2^*\}$ at exactly $\tau/2$ steps later, whether $w_1^*$ or $w_2^*$ will be visited is independent of which of $v_1^*$ or $v_2^*$ has been visited. Thus

$$\mathbb{P}(k(t) \geq d) \leq \mathbb{P}\left(\sum_{l \leq t} B_l \geq d\right)$$

$$= \mathbb{P}\left(\exp\left(\sum_{l \leq t} B_l\right) \geq e^d\right)$$

$$\leq e^{-d}\mathbb{E}[e^{\sum_{l \leq t} B_l}]$$

$$\leq e^{-d}\mathbb{E}[e^{\sum_{i=1}^{\lceil 2t/\tau \rceil} z_i}]$$

$$= e^{-d}(1 - q + eq)^{\lceil 2t/\tau \rceil}$$

$$\leq e^{\lceil 4t/\tau \rceil q - d}$$

Therefore, we conclude that for any $\delta \in (0, 1)$ and $q \in (0, 1/2)$ with probability at least $1 - \delta$,

$$k(t) < d, \quad \forall t \leq \frac{\tau(d - \log(1/\delta))}{4q}$$

which completes the proof of Lemma 5.2.

*Proof of Lemma 5.4:* To see the part of the smooth setting of Lemma 5.4, note that for any algorithm $\mathcal{A}$ with $B \geq 1$, we simply observe that by the construction of $F^*$ and $g^*$ and the Markov chain $P^*$, for any $t \geq 0$ and $m \in [M]$, there exists an algorithm $\tilde{\mathcal{A}}$ with $B = 1$ for which $\text{prog}_0(x_{t,m}^{\mathcal{A}[O_{F^*}]}) \leq \text{prog}_0(x_{t,m}^{\tilde{\mathcal{A}}[O_{F^*}]})$, since multiple samples do not contribute to additional progress of $x$, which then proves the part of smooth setting (Similar claims can be achieved for the mean-squared setting of Lemma 5.4).

*Proof of the smooth setting of Theorem 3.1:* Now to show the lower bound for the smooth setting of Theorem 3.1, setting

$$d = \min\left\{ \left\lfloor \frac{L\Delta}{4l_1 \Delta_0 \epsilon^2} \right\rfloor, \left\lfloor \frac{\sigma^2}{8a_1 \epsilon^2} \right\rfloor \right\} \tag{14}$$

and

$$\frac{1}{q} = 1 + \frac{\sigma^2}{8a_2 \epsilon^2} \tag{15}$$

yields that $F^* \in \mathcal{F}(\Delta, L)$ and Assumption 2.3 is satisfied. By Part (4) of Lemma A.1 and Lemma 5.2, choosing $\delta = 1/2$ renders that for any $m \in [M]$ with probability at least $1/2$,

$$\|\nabla F^*(x_{t,m}^{\mathcal{A}[O_{F^*}]})\| \geq 2\epsilon, \quad \forall t \leq \frac{\tau(d - 1)}{4q}$$

which implies that

$$\mathbb{E}\|\nabla F^*(x_{t,m}^{\mathcal{A}[O_{F^*}]})\| \geq \epsilon, \quad \forall t \leq \frac{\tau(d - 1)}{4q}.$$

Therefore, we conclude that

$$N_s^\epsilon(M, \Delta, L, \sigma^2, \tau) \geq \frac{\tau(d - 1)}{4q} \succeq \frac{\tau L\Delta}{\epsilon^2} + \frac{\tau\sigma^2}{\epsilon^4} \min\left\{ c_1 \sigma^2, c_2 L\Delta \right\}$$

by the selections of $d, q$ as (14),(15) for some constants $c_1, c_2 > 0$.

## C   Lower Bound for the Mean-squared Smooth Setting

In this section, we show the lower bound of the mean-squared smooth setting in Theorem 3.1. The idea is similar to the proof for the smooth setting, except that we replace the indicator function in (12) by its smoothed surrogate:

$$\Theta_i(x) := \Gamma\left( 1 - \left( \sum_{k=i}^{d} \Gamma^2(|x_k|) \right)^{1/2} \right) \tag{16}$$

where $\Gamma$ is defined by

$$\Gamma(t) = \frac{\int_{1/4}^{t} \Delta(\tau) d\tau}{\int_{1/4}^{1/2} \Delta(\tau) d\tau}$$

with

$$\Delta(t) = \begin{cases} 0, & t \leq 1/4 \text{ or } t \geq 1/2 \\ \exp\left(1/(100(t - 1/4)(t - 1/2))\right), & 1/4 < t < 1/2. \end{cases}$$

Then, we consider the following stochastic gradient $\bar{g}$:

$$\text{if } s \in \{v_1^*, v_2^*\}, \quad [\bar{g}(x; s)]_i = [\nabla h_1(x)]_i \cdot \left( 1 + \Theta_i(x) \left( \frac{\mathbb{1}_{s=v_1^*}}{q} - 1 \right) \right),$$

$$\text{if } s \in \{w_1^*, w_2^*\}, \quad [\bar{g}(x; s)]_i = [\nabla h_2(x)]_i \cdot \left( 1 + \Theta_i(x) \left( \frac{\mathbb{1}_{s=w_1^*}}{q} - 1 \right) \right),$$

$$\text{otherwise, } \bar{g}(x; s) = 0. \tag{17}$$

It is straightforward to see $\mathbb{E}_{s\sim\pi}[\bar{g}(x;s)] = \nabla F(x)$. According to Observation 1 of [5], we know $\Theta_i(x) = 0, \forall i \leq \text{prog}_{\frac{1}{2}}(x)$ and hence $[\bar{g}(x;s)]_i = [\nabla h_j(x)]_i, j = 1, 2, \forall i \leq \text{prog}_{\frac{1}{2}}(x)$ when $s \in \{v_1^*, v_2^*, w_1^*, w_2^*\}$. Moreover, according to Part (6) of Lemma A.1, we have for any $s \in \mathcal{S}$

$$[\bar{g}(x;s)]_i = 0, \quad \forall i > 1 + \text{prog}_{\frac{1}{2}}(x).$$

Defining $\delta(x;s) := \bar{g}(x;s) - \nabla F(x)$, it yields that there is eactly one non-zero coordinate of $\delta(x;s)$, which is the $1 + \text{prog}_{\frac{1}{2}}(x)$-th coordinate. Moreover, we have the following results for $\bar{g}$.

**Lemma C.1.** *Consider $\bar{g}$ defined as* (17). *Then,*

$$\mathbb{E}_{s\sim\pi}\|\bar{g}(x;s) - \nabla F(x)\|^2 \leq a_3 d + a_4 \frac{1-q}{q} \tag{18}$$

*for some constants $a_3, a_4 > 0$. And for some constant $\bar{l}_1 > 0$*

$$\mathbb{E}_{s\sim\pi}\|\bar{g}(x;s) - \bar{g}(y;s)\|^2 \leq \frac{\bar{l}_1^2}{q}\|x-y\|^2, \ \forall x, y \in \mathbb{R}^d. \tag{19}$$

*Proof.* Similar to the proof of Lemma B.2, we can easily obtain (18). To show (19), note that

$$\mathbb{E}\|\bar{g}(x;s) - \bar{g}(y;s)\|^2 = \mathbb{E}\|\delta(x;s) - \delta(y;s)\|^2 + \|\nabla F(x) - \nabla F(y)\|^2$$

$$= \sum_{i\in\{i_x^*, i_y^*\}} \mathbb{E}([\delta(x;s)]_i - [\delta(y;s)]_i)^2 + \|\nabla F(x) - \nabla F(y)\|^2$$

where $i_x^* = 1 + \text{prog}_{\frac{1}{2}}(x), i_y^* = 1 + \text{prog}_{\frac{1}{2}}(y)$. Since

$$\mathbb{E}([\delta(x;s)]_i - [\delta(y;s)]_i)^2 = ([\nabla F(x)]_i\Theta_i(x) - [\nabla F(y)]_i\Theta_i(y))^2 \frac{1-q}{q}$$

$$= ([\nabla F(x)]_i(\Theta_i(x) - \Theta_i(y)) + [\nabla F(x) - \nabla F(y)]_i\Theta_i(y))^2 \frac{1-q}{q}$$

$$\leq 2([\nabla F(x)]_i^2(\Theta_i(x) - \Theta_i(y))^2 + [\nabla F(x) - \nabla F(y)]_i^2\Theta_i(y)^2)\frac{1}{q}$$

and by Observation 1.3 of [5] $|\Theta_i(x) - \Theta_i(y)| \leq 36\|x-y\|$ and noting $|\Theta_i(x)| \leq 1, \|\nabla F(x)\|_\infty \leq 23$, we obtain

$$\mathbb{E}([\delta(x;s)]_i - [\delta(y;s)]_i)^2 \leq \frac{2}{q}(23^2 \cdot 36^2\|x-y\|^2 + \|\nabla F(x) - \nabla F(y)\|^2).$$

Finally leveraging Part (3) of Lemma A.1 gives

$$\mathbb{E}\|\bar{g}(x;s) - \bar{g}(y;s)\|^2 \leq \frac{\bar{l}_1^2}{q}\|x-y\|^2$$

with $\bar{l}_1^2 = 4 \cdot 23^2 \cdot 36^2 + 5l_1^2$. $\qquad\square$

Then we show the lower bound corresponding to the mean-squared smooth setting of Theorem 3.1.

*Proof of the mean-squared smooth setting of Theorem 3.1 and Lemma 5.3:* Noting that $\bar{L}$-mean-squared smoothness implies $\bar{L}$ smoothness, we thus consider the case $L \leq \bar{L}$ with $L$ to be determined later. Consider the same $F^*$ as (13) and let $\bar{g}^*(x;s) = (L\lambda/l_1)\bar{g}(x/\lambda;s)$. Similarly, we have $F^*$ is $L$-smooth. Also

$$\mathbb{E}_{s\sim\pi}\|\bar{g}^*(x;s) - \nabla F^*(x)\|^2 \leq \left(\frac{L\lambda}{l_1}\right)^2 (a_3 d + a_4(1-q)/q)$$

and

$$\mathbb{E}_{s\sim\pi}\|\bar{g}^*(x;s) - \bar{g}^*(y;s)\| = \left(\frac{L\lambda}{l_1}\right)^2 \mathbb{E}_{s\sim\pi}\|\bar{g}(x/\lambda;s) - \bar{g}(y/\lambda;s)\|^2$$

$$\leq \left(\frac{L\bar{l}_1}{l_1\sqrt{q}}\right)^2 \|x-y\|^2.$$

Then taking

$$d = \min\left\{ \left\lfloor \frac{L\Delta}{4l_1\Delta_0\epsilon^2} \right\rfloor, \left\lfloor \frac{\sigma^2}{8a_3\epsilon^2} \right\rfloor \right\}$$

$$\frac{1}{q} = \max\left\{ 1 + \frac{\sigma^2}{8a_4\epsilon^2}, \frac{\bar{l}_1^2}{l_1^2} \right\}$$

$$L = \frac{\bar{L}l_1\sqrt{q}}{\bar{l}_1} \leq \bar{L}$$

we guarantee that $F^* \in \mathcal{F}(L, \Delta)$ and Assumptions 2.3, 2.4 are satisfied. Similar to the proof of the smooth setting, we can easily obtain Lemma 5.3 and then conclude that

$$N^\epsilon(M, \Delta, \bar{L}^2, \sigma^2, \tau) \succeq \frac{\tau\bar{L}\Delta}{\epsilon} + \frac{\tau\bar{L}\Delta\sigma^2}{\epsilon^3}$$

which completes the proof.

# D   Convergence Analysis of MaC-SPIDER

In this section, we provide the proof for Section 4. We first present the following technical lemma.

**Lemma D.1.** *We have the following claims:*

- $d_{TV}(\mu P^{t+1}, \pi) \leq d_{TV}(\mu P^t, \pi)$.

- *For $k \geq 2$, $t_{mix}(2^{-k}) \leq (k-1)\tau_{mix}$.*

- *Moreover,*
$$\sum_{k=0}^{T} d_{TV}(\mu P^k, \pi) \leq 3\tau_{mix}, \quad \forall T \geq 0.$$

*Proof.* The first two claims are directly from [24].

To see the third claim, we note that

$$\sum_{k=0}^{T} d_{TV}(\mu P^k, \pi) \leq \sum_{k=0}^{\infty} d_{TV}(\mu P^k, \pi)$$

$$\leq \sum_{l=0}^{\tau_{mix}} d_{TV}(\mu P^l, \pi) + \sum_{k=0}^{\infty} \sum_{l=t_{mix}(2^{-k})+1}^{t_{mix}(2^{-(k+1)})} d_{TV}(\mu P^l, \pi)$$

$$\leq d_{TV}(\mu, \pi)\tau_{mix} + \sum_{k=2}^{\infty} (t_{mix}(2^{-(k+1)}) - t_{mix}(2^{-k}))2^{-k}$$

$$\leq d_{TV}(\mu, \pi)\tau_{mix} + \sum_{k=2}^{\infty} k2^{-k}\tau_{mix}$$

$$\leq d_{TV}(\mu, \pi)\tau_{mix} + 2\tau_{mix}$$

which completes the proof with $d_{TV}(\mu, \pi) + 2 \leq 3$. $\qquad\square$

## D.1   Proof of Lemma 4.1 and Proposition 4.3

*Proof of Lemma 4.1:* Let $\tilde{h}_t^i = h(s_{t+i}) - h_\pi$ and $\tilde{h}(s) = h(s) - h_\pi$. We have

$$\mathbb{E}_t \left\| \frac{1}{M} \sum_{i=1}^{M} h(s_{t+i}) - h_\pi \right\|^2 = \frac{1}{M^2} \sum_{i=1}^{M} \mathbb{E}_t \|\tilde{h}_t^i\|^2 + \frac{2}{M^2} \sum_{i=1}^{M-1} \sum_{j=i+1}^{M} \mathbb{E}_t \langle \tilde{h}_t^i, \tilde{h}_t^j \rangle.$$

First, we show the following useful bound: for any $s \in \mathcal{S}$, given $t \geq 0$ and $1 \leq i < j$,

$$\left( \sum_{s' \in \mathcal{S}} |P(s_{t+j} = s' \mid s_{t+i} = s) - \pi(s')| \|\tilde{h}(s')\| \right)^2$$

$$= \left( \sum_{s' \in \mathcal{S}} \pi(s')^{-1/2} |P(s_{t+j} = s' \mid s_{t+i} = s) - \pi(s')| \sqrt{\pi(s')} \|\tilde{h}(s')\| \right)^2$$

$$\leq \sum_{s' \in \mathcal{S}} \pi_{min}^{-1} |P(s_{t+j} = s' \mid s_{t+i} = s) - \pi(s')|^2 \sum_{s' \in \mathcal{S}} \pi(s') \|\tilde{h}(s')\|^2$$

$$\leq \left( \sum_{s' \in \mathcal{S}} \pi_{min}^{-1/2} |P(s_{t+j} = s' \mid s_{t+i} = s) - \pi(s')| \right)^2 \sigma^2$$

$$\leq 4\sigma^2 \pi_{min}^{-1} (\max_s d_{TV}(P^{j-i}(\cdot \mid s_t = s) - \pi))^2 \tag{20}$$

Then, noting that for any $1 \leq i < j \leq M$,

$$\mathbb{E}_t \langle \tilde{h}_t^i, \tilde{h}_t^j \rangle = \sum_{s \in \mathcal{S}} P(s_{t+i} = s \mid s_t) \sum_{s' \in \mathcal{S}} P(s_{t+j} = s' \mid s_{t+i} = s) \langle \tilde{h}(s), \tilde{h}(s') \rangle$$

$$= \sum_{s \in \mathcal{S}} (P(s_{t+i} = s \mid s_t) - \pi(s)) \sum_{s' \in \mathcal{S}} (P(s_{t+j} = s' \mid s_{t+i} = s) - \pi(s')) \langle \tilde{h}(s), \tilde{h}(s') \rangle$$

$$+ \sum_{s \in \mathcal{S}} \pi(s) \sum_{s' \in \mathcal{S}} (P(s_{t+j} = s' \mid s_{t+i} = s) - \pi(s')) \langle \tilde{h}(s), \tilde{h}(s') \rangle$$

$$+ \sum_{s \in \mathcal{S}} P(s_{t+i} = s \mid s_t) \sum_{s' \in \mathcal{S}} \pi(s') \langle \tilde{h}(s), \tilde{h}(s') \rangle$$

$$\leq \sum_{s \in \mathcal{S}} |P(s_{t+i} = s \mid s_t) - \pi(s)| \sum_{s' \in \mathcal{S}} |P(s_{t+j} = s' \mid s_{t+i} = s) - \pi(s')| \|\tilde{h}(s)\| \|\tilde{h}(s')\|$$

$$+ \sum_{s \in \mathcal{S}} \pi(s) \sum_{s' \in \mathcal{S}} |P(s_{t+j} = s' \mid s_{t+i} = s) - \pi(s')| \|\tilde{h}(s)\| \|\tilde{h}(s')\|$$

$$\leq 4\sigma^2 \pi_{min}^{-1} \max_s d_{TV}(P^i(\cdot \mid s), \pi) \cdot \max_s d_{TV}(P^{j-i}(\cdot \mid s), \pi)$$

$$+ 2\sigma^2 \pi_{min}^{-1/2} \max_s d_{TV}(P^{j-i}(\cdot \mid s), \pi)$$

where we use (20) and note $\sum_{s'} \pi(s') \langle \tilde{h}(s), \tilde{h}(s') \rangle = \langle \tilde{h}(s), \sum_{s'} \pi(s') \tilde{h}(s') \rangle = \langle \tilde{h}(s), \mathbb{E}_{s' \sim \pi}[\tilde{h}(s')] \rangle = 0$. Further using the fact $\sum_{t=1}^T \max_s d_{TV}(P^t(\cdot \mid s), \pi) \leq 3\tau_{mix}$ by Lemma D.1, we obtain

$$\frac{2}{M^2} \sum_{i=1}^{M-1} \sum_{j=i+1}^{M} \mathbb{E}_t \langle \tilde{h}_t^i, \tilde{h}_t^j \rangle \leq \frac{72 \tau_{mix}^2 \sigma^2}{\pi_{min} M^2} + \frac{6 \tau_{mix} \sigma^2}{\sqrt{\pi_{min}} M}.$$

Similarly we note that

$$\mathbb{E}_t \|\tilde{h}_t^i\|^2 = \sum_{s \in \mathcal{S}} P(s_{t+i} = s \mid s_t) \|\tilde{h}(s)\|^2$$

$$= \sum_{s \in \mathcal{S}} (P(s_{t+i} = s \mid s_t) - \pi(s)) \|\tilde{h}(s)\|^2 + \sum_{s \in \mathcal{S}} \pi(s) \|\tilde{h}(s)\|^2$$

$$\leq \sum_{s \in \mathcal{S}} |P(s_{t+i} = s \mid s_t) - \pi(s)| \|\tilde{h}(s)\|^2 + \sigma^2$$

$$\leq \pi_{min}^{-1} \sum_{s \in \mathcal{S}} |P(s_{t+i} = s \mid s_t) - \pi(s)| \sum_{s \in \mathcal{S}} \pi(s) \|\tilde{h}(s)\|^2 + \sigma^2$$

$$\leq 2\sigma^2 \pi_{min}^{-1} \max_s d_{TV}(P^i(\cdot \mid s_t), \pi) + \sigma^2$$

which then implies by Lemma D.1

$$\frac{1}{M^2} \sum_{i=1}^{M} \mathbb{E}_t \|\tilde{h}_t^i\|^2 \leq \frac{\sigma^2}{M} + \frac{6\tau_{mix}\sigma^2}{\pi_{min}M^2}.$$

Combining all above gives (7).

To show (9), we note that

$$
\mathbb{E}_t \left\| \frac{1}{M} \sum_{i=1}^{M} h(s_{t+i}) - h_\pi \right\|^2
$$

$$
= \mathbb{E}_t \left\| \frac{1}{M} \sum_{i=1}^{M} h(s_{t+i}) - \mathbb{E}_t \left( \frac{1}{M} \sum_{i=1}^{M} h(s_{t+i}) \right) + \mathbb{E}_t \left( \frac{1}{M} \sum_{i=1}^{M} (h(s_{t+i}) - h_\pi) \right) \right\|^2
$$

$$
\leq 2\mathbb{E}_t \left\| \frac{1}{M} \sum_{i=1}^{M} h(s_{t+i}) \right\|^2 + 2 \left\| \mathbb{E}_t \left( \frac{1}{M} \sum_{i=1}^{M} (h(s_{t+i}) - h_\pi) \right) \right\|^2
$$

where we use $(a + b)^2 \leq 2a^2 + 2b^2$ and $\mathbb{E}[X - \mathbb{E}(X)]^2 \leq \mathbb{E}[X]^2$.

We note that by replacing $\tilde{h}(s)$ in (20) by $h(s)$,

$$
\left\| \mathbb{E}_t \left( \frac{1}{M} \sum_{i=1}^{M} (h(s_{t+i}) - h_\pi) \right) \right\| = \left\| \frac{1}{M} \sum_{i=1}^{M} \sum_{s \in \mathcal{S}} (P(s_{t+i} = s \mid s_t) - \pi(s)) h(s) \right\|
$$

$$
\leq \frac{1}{M} \sum_{i=1}^{M} \sum_{s \in \mathcal{S}} |P(s_{t+i} = s \mid s_t) - \pi(s)| \|h(s)\|
$$

$$
\leq \frac{2B}{\sqrt{\pi_{min}}M} \sum_{i=1}^{M} \max_s d_{TV}(P^i(\cdot \mid s), \pi)
$$

$$
\leq \frac{6\tau_{mix}B}{\sqrt{\pi_{min}}M}
$$

where we use $\mathbb{E}_{s\sim\pi} \|h(s)\| \leq (\mathbb{E}_{s\sim\pi}\|h(s)\|^2)^{1/2} \leq B$ and this concludes (8). Moreover, similar to the analysis of $\mathbb{E}_t \langle \tilde{h}_t^i, \tilde{h}_t^j \rangle$, we have

$$
\mathbb{E}_t \left\| \frac{1}{M} \sum_{i=1}^{M} h(s_{t+i}) \right\|^2 = \frac{B^2}{M} + \frac{6\tau_{mix}B^2}{\pi_{min}M^2} + \frac{2}{M} \sum_{i=1}^{M-1} \sum_{j=i+1}^{M} \mathbb{E}_t \langle h(s_{t+i}), h(s_{t+j}) \rangle
$$

$$
\leq \frac{B^2}{M} + \frac{6\tau_{mix}B^2}{\pi_{min}M^2} + \frac{6\tau_{mix}B^2}{\sqrt{\pi_{min}}M} + \frac{72\tau_{mix}^2 B^2}{\pi_{min}M^2}
$$

$$
\leq \frac{7\tau_{mix}B^2}{\sqrt{\pi_{min}}M} + \frac{78\tau_{mix}^2 B^2}{\pi_{min}M^2}
$$

and hence

$$
\mathbb{E}_t \left\| \frac{1}{M} \sum_{i=1}^{M} h(s_{t+i}) - h_\pi \right\|^2 \leq \frac{14\tau_{mix}B^2}{\sqrt{\pi_{min}}M} + \frac{228\tau_{mix}^2 B^2}{\pi_{min}M^2}
$$

which concludes (9).

*Proof of Proposition 4.3:* We denote $\mathbb{E}_t(\cdot)$ as the expectation conditioning on filtration $\mathcal{F}_t$.

Note that for $t \bmod r = 0$,

$$\mathbb{E}_t \|v_t - \nabla F(x_t)\|^2 = \mathbb{E}_t \left\| \frac{1}{M_1} \sum_{i=1}^{M_1} g(x_t; s_{N_t+i}) - \nabla F(x_t) \right\|^2$$

$$\leq \frac{7\tau_{mix}\sigma^2}{\sqrt{\pi_{min}}M_1} + \frac{78\tau_{mix}^2\sigma^2}{\pi_{min}M_1^2}$$

$$\leq \frac{\epsilon^2}{8}$$

by (7) of Lemma 4.1 and noting $M_1 = 112\tau_{mix}\pi_{min}^{-1/2}\epsilon^{-2}\max\{\sigma, \sigma^2\}$.

For $t \bmod r \neq 0$, conditioning on $\mathcal{F}_{t+1}$ and letting $\tilde{g}_{t+1}^{M_2} = \frac{1}{M_2}\sum_{i=1}^{M_2} g(x_{t+1}; s_{N_{t+1}+i}) - g(x_t; s_{N_{t+1}+i})$ yields

$$\mathbb{E}_{t+1}\|v_{t+1} - \nabla F(x_{t+1})\|^2 = \mathbb{E}_{t+1}\left\| v_t - \nabla F(x_t) + \tilde{g}_{t+1}^{M_2} - \nabla F(x_{t+1}) + \nabla F(x_t) \right\|^2$$

$$= \|v_t - \nabla F(x_t)\|^2 + \mathbb{E}_{t+1}\|\tilde{g}_{t+1}^{M_2} - \nabla F(x_{t+1}) + \nabla F(x_t)\|^2$$

$$+ 2\langle v_t - \nabla F(x_t), \mathbb{E}_{t+1}[\tilde{g}_{t+1}^{M_2} - \nabla F(x_{t+1}) + \nabla F(x_t)]\rangle$$

$$\leq \|v_t - \nabla F(x_t)\|^2 + \mathbb{E}_{t+1}\|\tilde{g}_{t+1}^{M_2} - \nabla F(x_{t+1}) + \nabla F(x_t)\|^2$$

$$+ 2\|v_t - \nabla F(x_t)\|\|\mathbb{E}_{t+1}[\tilde{g}_{t+1}^{M_2} - \nabla F(x_{t+1}) + \nabla F(x_t)]\|$$

Noting that $\mathbb{E}_{t+1, s\sim\pi}\|g(x_{t+1}; s) - g(x_t; s)\| \leq L\|x_{t+1} - x_t\|, \forall s \in \mathcal{S}$ and $\mathbb{E}_{t+1, s\sim\pi}[g(x_{t+1}; s) - g(x_t; s)] = \nabla F(x_{t+1}) - \nabla F(x_t)$, combining with (8) and (9) of Lemma 4.1 gives

$$\mathbb{E}_{t+1}\|v_{t+1} - \nabla F(x_{t+1})\|^2 \leq \|v_t - \nabla F(x_t)\|^2 + \frac{12\tau_{mix}B}{\sqrt{\pi_{min}}M_2}\|v_t - \nabla F(x_t)\| + \frac{14\tau_{mix}B^2}{\sqrt{\pi_{min}}M_2} + \frac{228\tau_{mix}^2 B^2}{\pi_{min}M_2^2}.$$

where $B := L\max_t \|x_{t+1} - x_t\|$. Further noting

$$M_2 = \frac{16\tau_{mix}}{\sqrt{\pi_{min}}\epsilon}, \quad B \leq L\max_t\{\eta_t\|v_t\|\} \leq \frac{\epsilon}{4}$$

we have

$$\mathbb{E}_{t+1}\|v_{t+1} - \nabla F(x_{t+1})\|^2 \leq \|v_t - \nabla F(x_t)\|^2 + \frac{\epsilon^2}{4}\|v_t - \nabla F(x_t)\| + \frac{\epsilon^3}{16} + \frac{\epsilon^4}{16}$$

$$\leq \|v_t - \nabla F(x_t)\|^2 + \frac{\epsilon}{2}\|v_t - \nabla F(x_t)\|^2 + \frac{1}{8}(\epsilon^3 + \epsilon^4/2)$$

where we use $\frac{\epsilon^2}{4}\|v_t - \nabla F(x_t)\| \leq \frac{\epsilon}{2}\|v_t - \nabla F(x_t)\|^2 + \frac{1}{2\epsilon}(\frac{\epsilon^2}{4})^2$ in the second inequality. Then noting that for $rt_0 \leq t < r(t_0 + 1)$ given any $t_0 \geq 0$, we have for $r = 1/\epsilon$

$$\mathbb{E}\|v_t - \nabla F(x_t)\|^2 \leq (1 + \epsilon/2)^r \mathbb{E}\|v_{rt_0} - \nabla F(x_{rt_0})\|^2 + (1 + \epsilon/2)^r(\epsilon^3 + \epsilon^4/2)\cdot\frac{1}{4\epsilon}$$

$$\leq 2\cdot\frac{\epsilon^2}{8} + \frac{\epsilon^2}{2} + \frac{\epsilon^3}{4}$$

$$\leq \frac{3\epsilon^2}{4} + \frac{\epsilon^3}{4}$$

where we use the fact $(1 + \epsilon/2)^{1/\epsilon} \leq \sqrt{e} \leq 2$ and $\mathbb{E}\|v_{rt_0} - \nabla F(x_{rt_0})\|^2 \leq \epsilon^2/8$.

### D.2 Proof of Theorem 4.4

Noting $\bar{L}$-mean-squared smoothness implies $\bar{L}$-smoothness of $F$, we have

$$F(x_{t+1}) - F(x_t) \leq \langle \nabla F(x_t), x_{t+1} - x_t\rangle + \frac{\bar{L}}{2}\|x_{t+1} - x_t\|^2$$

$$= -\eta_t\langle\nabla F(x_t), v_t\rangle + \frac{\bar{L}\eta_t^2}{2}\|v_t\|^2$$

$$= -\eta_t\langle\nabla F(x_t) - v_t, v_t\rangle + \frac{\bar{L}\eta_t^2}{2}\|v_t\|^2 - \eta_t\|v_t\|^2$$

$$\leq -\frac{\eta_t}{2}\left(1 - \bar{L}\eta_t\right)\|v_t\|^2 + \frac{\eta_t}{2}\|v_t - \nabla F(x_t)\|^2.$$

Noting that $\eta_t \leq \frac{1}{2L}$, then using the fact that $\min\{|x|, x^2/2\} \geq |x| - 2, \forall x \in \mathbb{R}$,

$$\frac{\eta_t}{2}(1 - \bar{L}\eta_t)\|v_t\|^2 \geq \frac{\eta_t}{4}\|v_t\|^2$$
$$= \frac{\epsilon^2}{16L} \min\left\{\frac{\|v_t\|^2}{2\epsilon^2}, \frac{\|v_t\|}{\epsilon}\right\}$$
$$\geq \frac{\epsilon^2}{16L}\left(\frac{\|v_t\|}{\epsilon} - 2\right)$$
$$= \frac{\epsilon}{16L}\|v_t\| - \frac{\epsilon^2}{8L}$$

which hence induces

$$F(x_{t+1}) - F(x_t) \leq -\frac{\epsilon}{16\bar{L}}\|v_t\| + \frac{\epsilon^2}{8\bar{L}} + \frac{1}{4\bar{L}}\|v_t - \nabla F(x_t)\|^2.$$

Taking expectation on both sides and using Lemma 4.3 gives

$$\frac{\epsilon}{16\bar{L}}\mathbb{E}\|v_t\| \leq \mathbb{E}[F(x_t) - F(x_{t+1})] + \frac{3\epsilon^2}{8\bar{L}}$$

which indicates

$$\frac{1}{T}\sum_{t=0}^{T-1}\mathbb{E}\|v_t\| \leq \frac{16\bar{L}\mathbb{E}[F(x_0) - F(x_T)]}{T\epsilon} + 6\epsilon$$
$$\leq \frac{16\bar{L}\Delta_0}{T\epsilon} + 6\epsilon.$$

By $T = 16\bar{L}\Delta_0\epsilon^{-2}$ we conclude

$$\mathbb{E}\|\nabla F(\tilde{x}_T)\| = \frac{1}{T}\sum_{t=0}^{T-1}\mathbb{E}\|\nabla F(x_t)\|$$
$$\leq \frac{1}{T}\sum_{t=0}^{T-1}(\mathbb{E}\|v_t\| + \mathbb{E}\|v_t - \nabla F(x_t)\|)$$
$$\leq 7\epsilon.$$

It is straightforward to see that the total number of samples is upper bounded by

$$\left\lceil\frac{T}{r}\right\rceil(M_1 + M_2r) = \mathcal{O}(\tau_{mix}\pi_{min}^{-1/2}\epsilon^{-3}).$$

