# OpenReview forum: "Learning from A Single Markovian Trajectory: Optimality and Variance Reduction"
_NeurIPS.cc/2025/Conference — NeurIPS 2025 poster_

### Official Review · Reviewer_2E35 · 2025-06-27

**Clarity:** 4
**Significance:** 4
**Originality:** 4
**Rating:** 4
**Confidence:** 1

**Summary:**

This paper explores a general stochastic non-convex optimization problem, which lies outside the scope of my current research expertise and academic focus. As such, I do not feel adequately qualified to provide a fair and informed evaluation of the technical contributions and significance of the work. I respectfully request that my review be disregarded in the decision-making process, and that greater consideration be given to the assessments provided by reviewers with relevant expertise in this area.

**Questions:**

This paper explores a general stochastic non-convex optimization problem, which lies outside the scope of my current research expertise and academic focus. As such, I do not feel adequately qualified to provide a fair and informed evaluation of the technical contributions and significance of the work. I respectfully request that my review be disregarded in the decision-making process, and that greater consideration be given to the assessments provided by reviewers with relevant expertise in this area.

**Ethical Concerns:**

["NO or VERY MINOR ethics concerns only"]

**Final Justification:**

This paper is out of the scope of my research area. Please ignore my review and consider the reviews from other reviewers. Please find other reviewers and assign this paper to them.

**Limitations:**

yes

**Quality:**

4

**Strengths And Weaknesses:**

This paper explores a general stochastic non-convex optimization problem, which lies outside the scope of my current research expertise and academic focus. As such, I do not feel adequately qualified to provide a fair and informed evaluation of the technical contributions and significance of the work. I respectfully request that my review be disregarded in the decision-making process, and that greater consideration be given to the assessments provided by reviewers with relevant expertise in this area.

---

> ### Author Rebuttal · Authors · 2025-07-31
>
> We appreciate the reviewer for the kind review.

---

### Official Review · Reviewer_obPR · 2025-06-30

**Clarity:** 2
**Significance:** 3
**Originality:** 3
**Rating:** 4
**Confidence:** 1

**Summary:**

This paper studies stochastic non-convex optimization with samples from a single uncontrolled Markov chain. It establishes an $\Omega(\epsilon^{-3})$ lower bound under the mean-squared smoothness assumption and an $\Omega(\epsilon^{-4})$ bound without it. The proposed MaC-SPIDER algorithm matches these bounds, achieving near-optimal sample complexity.

**Questions:**

(1) In Remark 4.5, we see that there is still a gap between $\tau$ and $\pi_{min}^{1/2}\tau_{mix}$. Could you provide a concrete example to show this gap?

(2) Could you provide any empirical validation?

(3) What result does MaC-SPIDER have without the mean-squared smooth assumption? Can it also match the lower bound?

**Ethical Concerns:**

["NO or VERY MINOR ethics concerns only"]

**Final Justification:**

The rebuttal addresses my concerns. Thus, I lean towards acceptance.

**Limitations:**

Yes

**Quality:**

3

**Strengths And Weaknesses:**

Strengths:

The paper is well presented. It proposes a new algorithm based on the variance reduction (SPIDER) algorithm in optimization for the setting of learning from a single Markov chain. Lower bound analysis shows that the result is nearly minimax optimal.

Weaknesses:

(1) The abstract and the introduction are inconsistent: the lower bound in the abstract is $\epsilon^{-3}$, while that in the intro is $\epsilon^{-4}$.

(2) There are some typos in the sample complexity definition (equation 6, the left part should depend on both $\bar L$ and $L$). Moreover, it’s hard for me to understand the meaning of this metric. For example, it is not clear to me what the final infimum is taken over. Could you clarify more about its definition?

(3) The algorithm requires prior knowledge of some parameters like $\pi_{min}$ and $\tau_{\min}$, which is hard to know in advance.

---

> ### Author Rebuttal · Authors · 2025-07-31
>
> ## Response to Weakness 1
> We appreciate the reviewer for the comment. Actually, lower bounds of $\epsilon^{-3}$ and $\epsilon^{-4}$ are established under different settings, and hence there is no contradiction. For $\mathbf{\epsilon^{-4}}$, we only require $F$ to be **smooth**, while for $\mathbf{\epsilon^{-3}}$, we further need **mean-squared smoothness**, i.e., Assumption 2.4, which is also a widely-adopted assumption for variance reduction analysis of the i.i.d. case. Obviously, mean-squared smoothness indicates classical smoothness, meaning $\epsilon^{-3}$ lower bound is established under a smaller function class. Specifically, in the statement of Theorem 3.1 we clarify different settings under which two lower bounds are proved. We will further clarify their differences in the final version to avoid any misunderstanding.
>
>
> ## Response to Weakness 2
> Yes, the left part of eq. (6) also depends on $L$ and we will add it in the final version. In terms of the complexity measure, we explain its meaning as follows: First, the final infimum means the minimum number of samples are drawn when the gradient norm of $F$ evaluated at all historical outputs of algorithm $\mathcal{A}$ for the first time becomes less than $\epsilon$, i.e., optimality criterion is reached. Then, if we lower bound that measure by some $N$, then we basically conclude that there exist some $\pi$, some $F, g$ (due to supremum) such that for any algorithm (due to infimum), the minimum number of samples to be drawn for the algorithm to achieve arbitrarily small gradient norm for the first time is at least $N$. This establishes the meaning of algorithm-independent sample complexity lower bound. See Lines 185-191 for discussions as well.
>
>
> ## Response to Weakness 3
> We acknowledge that the implementation of $\pi_min$ and $\tau_{mix}$. However, we note that for existing algorithms in literature, the knowledge of these problem-dependent quantities are unavoidable for algorithm design.  For example, for vanilla SGD, the learning rate is chosen as $\mathcal{O}(1/(\tau_{mix}\log T))$ [Even’23]; for Randomized GD, the batch size is set as $\mathcal{O}(\tau_{mix}\sqrt{T})$ [Beznosikov et al’24]. It is also worth noting that convergence results of the above-mentioned algorithms are established on a stronger assumption $\Vert g(x;s) - \nabla F(x) \Vert \le \sigma, \forall s$ rather than the one we consider in this paper (see Assumption 2.3). In fact, when assuming $\Vert g(x;s) - \nabla F(x) \Vert \le \sigma, \forall s$, we can get rid of the dependence of $\pi_{min}$ in Theorem 4.4. This guarantees that **our algorithm needs exactly the same knowledge of problem-dependent quantities as in literature**. Furthermore, even under Assumption 2.3, we have a practical way to estimate $\pi_{min}$. Specifically, the idea is to maintain a vector $y \in \mathbb{R}^{|\mathcal{S}|}$, where $y(s) = \sum_{i=1}^{N} h(s_i) / N$ with $h$ being the indicator function, and N being the number of total samples. By Lemma 4.1, it is straightforward to obtain $y \to \pi$ as $N \to \infty$ and hence $\min y(s) \to \pi_{min}$. Therefore, by utilizing estimate $y$, we can get rid of knowing $\pi_{min}$.
>
>
> ## Response to Question 1
> For example, if we consider directed cycle with self-loop, i.e., for state $i$, it has probability $1/2$ staying at the current state or transiting to $i+1$ mod $n$, where $n$ is the number of states, the hitting time is $2n$. The mixing time of this chain is $\Theta(n^2)$ and $\pi_i = 1/n, \forall i$. Therefore, the gap is $\Theta(n^{3/2})$.
>
>
> ## Response to Question 2
> We appreciate the reviewer for the suggestion on empirical validation. We have added the empirical comparison between SGD and MaC-SPIDER with Markov sampling. And we are willing to add more baselines in the final version.
>
> In particular, we have run experiments under MNIST dataset with a customized Markov sampler. Each data point in the training split of MNIST is treated as a unique state, i.e., there are in total 60k states. The Markov sampler is defined to be sparse such that given the $i$-th data point is sampled, ten of the remaining data points have equal probability to be sampled in the next step of Markov chain evolution. We set $\epsilon=0.05$ for both SGD and MaC-SPIDER. Then we choose the batch size of SGD by 500 (which is ~ $\epsilon^{-2}$ as in literature) and let $M_1 = 500$ (~ $\epsilon^{-2}$), $M_2 = 20$ (~$\epsilon^{-1}$) as we indicate in the paper. For the learning rates, we set 0.02 for SGD and 0.5 for MaC-SPIDER. Our experimental results show that with the same amount of drawn samples, MaC-SPIDER achieves better performance than SGD. In other words, MaC-SPIDER has lower sample complexity than SGD. See the following table for detail. Moreover, as shown in the table, **with only 50k samples, MaC-SPIDER has achieved comparable performance as SGD with 90k samples**, which further indicates its sample efficiency and faster convergence.
>
> | Performance | SGD (50k samples) | MaC-SPIDER (50k samples) |SGD (90k samples) | MaC-SPIDER (90k samples) |
> |----------------|------------------|------------------|------------------|------------------|
> | Train Loss   | 0.8068    | 0.3454     | 0.3776    | 0.2691     |
> | Test Loss   | 0.7931     | 0.3305     | 0.3653     | 0.2571     |
> | Train Acc (%)   | 71.43| 89.53     | 89.01     | 91.91     |
> | Test Acc (%)   | 71.57     | **90.34**     | 89.49     | **92.47**     |
>
>
> ## Response to Question 3
> Without assuming mean-squared smoothness, the upper bound of MaC-SPIDER becomes $\tau_{mix} \pi_{min} \epsilon^{-4}$. We note that this $\epsilon^{-4}$ result matches the first lower bound in Theorem 3.1, since now only smoothness assumption of $F$ is placed, i.e., MaC-SPIDER is still nearly min-max optimal for smooth (but not mean-squared smooth) functions. This also guarantees that without mean-squared smoothness, it is impossible for MaC-SPIDER to achieve $\epsilon^{-3}$ complexity. In fact, mean-squared smoothness is a common assumption to achieve $\epsilon^{-3}$ complexity in variance reduction literature for the i.i.d. case.
>
>
> **References**
>
> [1]. Mathieu Even. Stochastic gradient descent under markovian sampling schemes. In International Conference on Machine Learning, pages 9412–9439. PMLR, 2023.
>
> [2]. Aleksandr Beznosikov, Sergey Samsonov, Marina Sheshukova, Alexander Gasnikov, Alexey Naumov, and Eric Moulines. First order methods with markovian noise: from acceleration to variational inequalities. Advances in Neural Information Processing Systems, 36, 2024.

---

> > ### Comment · Reviewer_obPR · 2025-08-01
> > **Response to the authors**
> >
> > Thanks for your clarification.
> >
> > Weakness 1: I understand the results are obtained with different assumptions. My key point is that the abstract does not mention the assumption. Thus, the claim should be further clarified to avoid any confusion.
> >
> > I don't have any other concerns after reading the authors' rebuttal. I will keep my score and lean towards acceptance.

---

> ### Author Response · Authors · 2025-08-01
>
> Thank you for the suggestion! We will clarify the assumption in the updated abstract. And we really appreciate your positive feedback and support.

---

### Official Review · Reviewer_7A1K · 2025-07-03

**Clarity:** 3
**Significance:** 2
**Originality:** 2
**Rating:** 4
**Confidence:** 2

**Summary:**

This paper looks at optimizing non-convex functions when data comes from a single Markov chain trajectory—no resets, no i.i.d. samples. That makes things harder due to data correlation. The authors prove a lower bound showing you need at least Ω(ϵ⁻³) samples in the best case. Then they propose MaC-SPIDER, a smart variance-reduced method that hits this bound (up to constants), making it nearly optimal. It’s the first method to do this using just one trajectory. The work is purely theoretical and highlights that it doesn't cover infinite or non-stationary chains.

**Questions:**

N/A

**Ethical Concerns:**

["NO or VERY MINOR ethics concerns only"]

**Final Justification:**

Thanks for the clarification. I've raised my score.

**Quality:**

3

**Strengths And Weaknesses:**

Weakness:
1. The novelty is limited. The paper extends known SPIDER-style variance reduction to the Markov setting, but the core idea is a relatively straightforward adaptation. The analysis borrows heavily from prior i.i.d. literature and lower bound techniques.

2.The core technical difficulty (dealing with Markovian bias and variance) is acknowledged. The novelty is incremental.

3. The method is motivated by practical problems like RL, but the paper does not test on any real or synthetic Markovian datasets to show relevance.

4.The algorithm relies on mixing time and stationary distribution parameters, but how to estimate them or deal with their unknown nature?

5. The technique can reduce variance, but how to handle the bias introduced by Markovian sampling?

---

> ### Author Rebuttal · Authors · 2025-07-31
>
> ## Response to Weaknesses 1 and 2
> We would like to argue that analysis of MaC-SPIDER is technically non-trivial, even though it algorithmically looks similar to SPIDER. In fact, the correlation among Markovian samples introduces much more difficulty in analyzing variance control using $v_t$. Particularly, Lemma 4.1 is the core result that we leverage to reduce variance. In the i.i.d. case, it is straightforward to obtain
> $
> 	\mathbb{E}\left\Vert \frac{1}{M} \sum_{i=1}^M h(s_{t+i}) - h_{\pi} \right\Vert^2 = \frac{1}{M} \sum_{i=1}^{M} \mathbb{E}\Vert h(s_{t+i}) - h_{\pi} \Vert^2
> $
> due to independence across samples. However, for the Markovian case, this equation fails to hold. In particular, if simply using $\Vert \sum_{i=1}^M a_i \Vert^2 \le M \sum_{i=1}^M \Vert a_i \Vert^2$, we only have $\mathbb{E}\left\Vert \frac{1}{M} \sum_{i=1}^M h(s_{t+i}) - h_{\pi} \right\Vert^2 = \sum_{i=1}^{M} \mathbb{E}\Vert h(s_{t+i}) - h_{\pi} \Vert^2$, which makes the variance uncontrollable. Therefore, to obtain a tighter bound, we carefully analyze the correlation among Markov samples, combined further with the mixing theory of Markov chains, to finally obtain the results in Lemma 4.1. We would like to highlight again that deriving the variance bounds in Lemma 4.1 needs new mathematical tools that are not required for the i.i.d. setting. We refer the reviewer to Appendix D.1 for detailed proof.
>
> Besides much more difficulty in deriving Lemma 4.1, we further note that even with Lemma 4.1 ready to use, achieving $\mathbb{E}\Vert v_t - \nabla F(x_t) \Vert^2 \le \epsilon^2$ needs careful and technical analysis which is not required for the i.i.d. case. Specifically, in Line 898 in Appendix, the cross product term cannot be neglected (which is instead zero in the i.i.d. case), which thus induces more sophisticated analysis to achieve the final bound (see Lines 898-904 for more details). Therefore, we argue that the analysis is much more challenging and we indeed use new techniques and mathematical tools to deal with the challenge which is not encountered in the i.i.d. case.
>
> In terms of the lower bound analysis, we acknowledge that we borrow the idea of function construction in [Arjevani et al’23]. However, we note that naively adopting i.i.d. analysis fails to obtain the lower bounds that depend on $\tau$. In fact, our construction is more challenging in the following sense: Unlike the i.i.d. case considered in [Arjevani et al’23], where only unified, state-independent stochastic gradient $g$ are constructed, in our paper, due to inherent correlation across different states caused by Markovian property, we construct $g$ which is a function of states (see eq. (12) in Appendix B). Then a Markov chain is carefully designed (see Appendix A) such that combined with constructed $g$, making one increase in the number of non-zero elements (i.e., progress) at least takes $\tau$ iterations (while noting that for the i.i.d. case making progress at least takes only one iteration [Arjevani et al’23]). In particular, we note that bounding the probability of progress in Lines 840-844 is totally different from the i.i.d. case, as Markovian samples are correlated. This distinguishes our analysis from classical lower bound analysis for the i.i.d. case.
>
>
> ## Response to Weakness 3
> We appreciate the reviewer for the suggestion on adding empirical evaluation. We have added the empirical results of MaC-SPIDER and SGD with Markov sampling MNIST dataset. We would like to add more experiments for real-world RL problems in the final version due to the time limit.
>
> In particular, we have run experiments under MNIST dataset with a customized Markov sampler. Each data point in the training split of MNIST is treated as a unique state, i.e., there are in total 60k states. The Markov sampler is defined to be sparse such that given the $i$-th data point is sampled, ten of the remaining data points have equal probability to be sampled in the next step of Markov chain evolution. We set $\epsilon=0.05$ for both SGD and MaC-SPIDER. Then we choose the batch size of SGD by 500 (which is ~ $\epsilon^{-2}$ as in literature) and let $M_1 = 500$ (~ $\epsilon^{-2}$), $M_2 = 20$ (~$\epsilon^{-1}$) as we indicate in the paper. For the learning rates, we set 0.02 for SGD and 0.5 for MaC-SPIDER. Our experimental results show that with the same amount of drawn samples, MaC-SPIDER achieves better performance than SGD. In other words, MaC-SPIDER has lower sample complexity than SGD. See the following table for detail. Moreover, as shown in the table, **with only 50k samples, MaC-SPIDER has achieved comparable performance as SGD with 90k samples**, which further indicates its sample efficiency and faster convergence.
>
>
> | Performance | SGD (50k samples) | MaC-SPIDER (50k samples) |SGD (90k samples) | MaC-SPIDER (90k samples) |
> |----------------|------------------|------------------|------------------|------------------|
> | Train Loss   | 0.8068    | 0.3454     | 0.3776    | 0.2691     |
> | Test Loss   | 0.7931     | 0.3305     | 0.3653     | 0.2571     |
> | Train Acc (%)   | 71.43| 89.53     | 89.01     | 91.91     |
> | Test Acc (%)   | 71.57     | **90.34**     | 89.49     | **92.47**     |
>
>
>
> ## Response to Weakness 4
> We acknowledge that the implementation of our algorithm needs the information of the mixing time and the minimal element of $\pi$. We note that if the upper and lower bounds of $\tau_{mix}$ and $\pi_{min}$, respectively, are known, (meaning we can upper bound $\tau_{mix}\pi_{min}^{-1/2}$) we can replace them by their corresponding bounds, which will not hurt the performance too much. Theoretically, by doing this, the sample complexity scales constantly (i.e., independent of $\epsilon$) according to the bound of $\tau_{mix}\pi_{min}^{-1/2}$. Moreover, we argue that existing algorithms with Markov sampling require hyperparameters to be chosen with the knowledge of problem-dependent quantities. For example, for vanilla SGD, the learning rate is chosen as $\mathcal{O}(1/(\tau_{mix}\log T))$ [Even’23]; for Randomized GD, the batch size is set as $\mathcal{O}(\tau_{mix}\sqrt{T})$ [Beznosikov et al’24]. We further note that the convergence results of the above-mentioned two algorithms are placed under a stronger bounded noise assumption, i.e., $\Vert g(x;s) - \nabla F(x) \Vert \le \sigma$ rather than the bounded variance assumption $\mathbb{E}\Vert g(x;s) - \nabla F(x) \Vert^2 \le \sigma^2$ in our paper. If considering the same bounded noise assumption, our convergence results can further get rid of the dependence on $\pi_{min}$. That is to say, **our algorithm needs exactly the same knowledge of problem-dependent quantities as in literature**. Furthermore, when considering the bounded variance assumption (which is the focus of this paper), we present a simple way to estimate $\pi_{min}$, which can be plugged into MaC-SPIDER easily. The idea is to maintain a vector $y \in \mathbb{R}^{|\mathcal{S}|}$, where $y(s) = \sum_{i=1}^{N} h(s_i) / N$ with $h$ being the indicator function, and N being the number of total samples. By Lemma 4.1, it is straightforward to obtain $y \to \pi$ as $N \to \infty$ and hence $\min_s y(s) \to \pi_{min}$. Therefore, by plugging in estimate $y$, we can get rid of knowing $\pi_{min}$.
>
>
> ## Response to Weakness 5
> We note that although Markovian samples are biased, asymptotically there is no bias due to the mixing property of Markov chains. In Theorem 4.4, we show that for arbitrary small $\epsilon$, MaC-SPIDER can output a $\tilde{x}_T$ such that $\Vert \nabla F(\tilde{x}_T) \Vert \le 7\epsilon$, which indicates MaC-SPIDER has no bias while preserves variance reduction.
>
>
> **References**
>
> [1] Yossi Arjevani, Yair Carmon, John C Duchi, Dylan J Foster, Nathan Srebro, and Blake Wood333 worth. Lower bounds for non-convex stochastic optimization. Mathematical Programming, 199(1):165–214, 2023.
>
> [2] Aleksandr Beznosikov, Sergey Samsonov, Marina Sheshukova, Alexander Gasnikov, Alexey Naumov, and Eric Moulines. First order methods with markovian noise: from acceleration to variational inequalities. Advances in Neural Information Processing Systems, 36, 2024.

---

### Official Review · Reviewer_bozm · 2025-07-04

**Clarity:** 2
**Significance:** 3
**Originality:** 3
**Rating:** 5
**Confidence:** 2

**Summary:**

This work studies the problem of general stochastic non-convex optimization for $F(x)$ under the setting that the sampling process (randomness) follows a finite-state stationary Markov chain (MC), which has many applications in machine learning. It considers the case where only one trajectory is sampled and no restarts are allowed. The interaction protocol is through a stochastic first-order oracle, which returns unbiased and variance-bounded samples of the gradient ($\mathbb{E} [g(x, s)] = \nabla F(x)$), and the random seed $s$ is from the MC. This work then focuses on zero-respecting algorithms, where at each step $t$, the evaluation point $x_t$ can only be taken from previous points with non-zero gradients. This is a broad algorithm class which stochastic gradient descent and randomized extragradient fall in. First, a sample complexity lower bound of $\Omega (\epsilon^{-3})$ is proven, which is the first in the literature. Then, an algorithm, MaC-SPIDER, built upon a previous algorithm SPIDER, is proposed, and is shown that its upper bound is also $O (\epsilon^{-3})$ up-to other factors such as the mixing time and the minimal stationary probability.

**Questions:**

Do you think the problem-dependent quantities of $\tau_{mix}$ and $\pi_{min}$ are necessary in the lower-bound (in the minimax sense)? Why is the specific value of $1/4$ chosen for the mixing time?

**Ethical Concerns:**

["NO or VERY MINOR ethics concerns only"]

**Final Justification:**

The authors' response addresses my concerns and improves the clarity of this paper. I'll keep my positve score.

**Limitations:**

As the authors admitted, the setting of finite-state stationary MCs is restrictive. Though the dependence of $\epsilon$ is tight, the necessity of other problem-dependent quantities is not discussed.

**Quality:**

3

**Strengths And Weaknesses:**

### Strengths

- **Important Problem:** This work advances the theoretical understanding of stochastic optimization, which is crucial for modern ML applications where i.i.d. assumptions often fail. The results could influence algorithm design in machine learning, both theoretically and practically.

- **Fundamental Contribution:** First to establish matching (the gap being only problem-dependent constants) lower and upper bounds for single-trajectory Markovian optimization, filling an important gap in the literature.

- **Technical Innovation:** Successfully adapts variance reduction to handle bias and temporal dependence in Markovian samples, which is non-trivial.


### Weaknesses

- **Lack of Intuitive Explanation:** Section 2.2 introduces many concepts, which may confuse readers not specialized in this problem. The authors did not provide enough explanations for these settings.

- **Limited Scope:** Restricts to finite-state Markov chains, excluding many practical scenarios with continuous or countably infinite state spaces. Restricts to stationary MCs, limiting applicability to non-stationary settings.

- **Lack of Numerical Simulation:** An experiment showing MaC-SPIDER is indeed converging faster than previous algorithms is missing.

---

> ### Author Rebuttal · Authors · 2025-07-31
>
> ## Response to "Lack of Intuitive Explanation"
> We thank the reviewer for pointing out the clarification problem of Section 2.2. We will add more explanation in the final version and we also briefly explain it here for clarity. From high-level, one can think of the algorithm class considered in Section 2.2 has the following key properties:
>
> (1). For each $x$, one can calculate the per-sample first-order information (i.e., the gradient $g(x;s_i)$) with an arbitrary batch size (i.e., B can be arbitrary and time-varying);
>
> (2). For each iteration $t$ of the algorithm, there can be multiple query updates in $x$. For example, SGD with momentum lies in the class, since
> $$
> x_{t+1,1} = \beta x_{t,1} + (1-\beta)g(x_{t,2};s_t), ~~~~ x_{t+1, 2} = x_{t, 2} - \eta x_{t+1, 1} .
> $$
> And MaC-SPIDER is another example, where we have $x_{t, 1} = v_t$ and $x_{t, 2} = x_t$.
>
>
> (3). The update of next iteration in $x$ can use all history information, including history of all sampled gradients.
>
> (4). The algorithm is zero-respecting, meaning that the support of $x_t$ lies the union of supports of historical sampled gradients. This is a generalization to the linear span setting [Even’23, Beznosikov et al’24]. Moreover, we would like to note that for lower bound analysis with i.i.d. samples, similar assumptions on zero-respecting algorithms are placed [Arjevani et al’23].
>
> We also show algorithms in Lines 138-142 which are exact examples fitting in the algorithm class we consider in Section 2.2.
>
>
>
>
> ## Response to "Limited Scope"
> We acknowledge finite state space and stationarity are the limitations of the paper. However, we would like to note that even for the finite-state, stationary case, to the best of our knowledge, this paper is the first work to establish an $\epsilon^{-3}$ algorithm-independent lower bound for mean-squared smooth functions under Markovian samples, and it is also the first work which provides an algorithm under the same assumptions as those for the lower bound that achieves $\epsilon^{-3}$ upper bound (up to some constant), showing its min-max optimality. We believe extending the current results to infinite-state and non-stationary cases is non-trivial, and we hope to put it in our future plan.
>
> ## Response to "Lack of Numerical Simulation"
> Due to time limitation, we have only compared MaC-SPIDER with SGD under Markov sampling setting. And we would like to add more baseline algorithms in the final version.
>
> In particular, we have run experiments under MNIST dataset with a customized Markov sampler. Each data point in the training split of MNIST is treated as a unique state, i.e., there are in total 60k states. The Markov sampler is defined to be sparse such that given the $i$-th data point is sampled, ten of the remaining data points have equal probability to be sampled in the next step of Markov chain evolution. We set $\epsilon=0.05$ for both SGD and MaC-SPIDER. Then we choose the batch size of SGD by 500 (which is ~ $\epsilon^{-2}$ as in literature) and let $M_1 = 500$ (~ $\epsilon^{-2}$), $M_2 = 20$ (~$\epsilon^{-1}$) as we indicate in the paper. For the learning rates, we set 0.02 for SGD and 0.5 for MaC-SPIDER. Our experimental results show that with the same amount of drawn samples, MaC-SPIDER achieves better performance than SGD. In other words, MaC-SPIDER has lower sample complexity than SGD. See the following table for detail. Moreover, as shown in the table, **with only 50k samples, MaC-SPIDER has achieved comparable performance as SGD with 90k samples**, which further indicates its sample efficiency and faster convergence.
>
>
>
> | Performance | SGD (50k samples) | MaC-SPIDER (50k samples) |SGD (90k samples) | MaC-SPIDER (90k samples) |
> |----------------|------------------|------------------|------------------|------------------|
> | Train Loss   | 0.8068    | 0.3454     | 0.3776    | 0.2691     |
> | Test Loss   | 0.7931     | 0.3305     | 0.3653     | 0.2571     |
> | Train Acc (%)   | 71.43| 89.53     | 89.01     | 91.91     |
> | Test Acc (%)   | 71.57     | **90.34**     | 89.49     | **92.47**     |
>
>
> ## Response to "Questions"
>  We appreciate the reviewer for the comment on problem-dependent quantities. From our current development of lower bound proof, we are unclear about whether lower bounds can be related to $\tau_{mix}$ or $\pi_{min}$ or both. In fact, relations among $\tau$, $\tau_{mix}$ and $\pi_{min}$ vary a lot for different Markov chains. We would like to investigate their relationships and hope to develop lower bounds characterized by $\tau_{mix}, \pi_{min}$ in the future.
>
> In terms of why choosing 1/4 for the mixing time, the reason follows it is conventionally adopted in Markov chain literature [Levin and Peres’17]. Choosing another value does not affect our results. In the second statement of Lemma D.1., it is shown that $t_{mix}(2^{-k}) \le (k-1)t_{mix}(1/4), \forall k \ge 2$, i.e., different values only scale $\tau_{mix}$ by some constant.
>
>
> References:
>
> [1]. Mathieu Even. Stochastic gradient descent under markovian sampling schemes. In International Conference on Machine Learning, pages 9412–9439. PMLR, 2023.
>
> [2]. Aleksandr Beznosikov, Sergey Samsonov, Marina Sheshukova, Alexander Gasnikov, Alexey Naumov, and Eric Moulines. First order methods with markovian noise: from acceleration to variational inequalities. Advances in Neural Information Processing Systems, 36, 2024.
>
> [3]. Yossi Arjevani, Yair Carmon, John C Duchi, Dylan J Foster, Nathan Srebro, and Blake Wood333 worth. Lower bounds for non-convex stochastic optimization. Mathematical Programming, 199(1):165–214, 2023.
>
> [4]. David A Levin and Yuval Peres. Markov chains and mixing times, volume 107. American Mathematical Soc., 2017.

---

> > ### Comment · Reviewer_bozm · 2025-08-07
> >
> > The authors' response addresses my concerns and improves the clarity of this paper. I'll keep my positve score.

---

### Note · Authors · 2025-08-13

We sincerely appreciate the AC and all reviewers' valuable time and efforts on evaluating our work.

Overall, we are glad that reviewers found our paper interesting and provided positive feedback. In the rebuttal, we have addressed reviewers' concerns in the following sense:

1. **Experimental results**: as suggested by all reviewers, we compared MaC-SPIDER and SGD with Markov sampling on MNIST dataset. Our results show that MaC-SPIDER outperforms SGD when using the same amount of samples. Moreover, with only consuming roughly half amount of samples that SGD needs, MaC-SPIDER can achieve even better performance.

2. **Novelty and technical difficulty**: we noted that although MaC-SPIDER looks algorithmically similar to SPIDER, our analysis is non-trivial. We clarified that the intrinsic correlation among Markovian samples make the tools work for i.i.d. case fail to hold, and hence we developed new mathematical tools to carefully analyze and control the variance of gradient estimate. In terms of our lower bounds, we note that simply adopting ideas from the i.i.d. case cannot reveal dependence on $\tau$. Rather, we designed stochastic gradient $g$ to be state-dependent, which is also related to the Markov chain structure. Finally, we would like to highlight that our lower bounds and MaC-SPIDER demonstrate optimality in sample complexity for Markov cases, which is not well-understood in literature.

3. **Problem-dependent quantities**: we clarified that MaC-SPIDER needs exactly same amount of knowledge as in literature when considering the same bounded noise assumption. By generalizing to the bounded variance assumption (which is not studied in literature), we introduce $\pi_{min}$. However, in the rebuttal, we explained in practice how we could further estimate $\pi_{min}$.

4. **Clarification**: as pointed by the reviewers, Section 2.2 needs further explanations of some mathematical concepts. We provided clear explanations and discussions in the rebuttal.

In summary, in this paper, we provide the novel algorithm-independent lower bounds for any first-order algorithms of stochastic non-convex optimization problems with Markov sampling. And we propose a novel min-max optimal algorithm which only uses a single trajectory to achieve matched sample complexity as our lower bounds, showing what the best first-order algorithms can do in the Markovian cases.

---

### Decision · Program_Chairs · 2025-09-17

**Decision:**

Accept (poster)

**Comment:**

The paper studies stochastic non-convex optimization when samples are generated by a finite-state stationary Markov chain (MC). The authors derive lower bounds on the sample complexity for this setting, then adapt the SPIDER algorithm and show that the resulting method, MaC-SPIDER, achieves performance close to the lower bound, up to terms involving the mixing time and the minimal state visitation probability.

The reviewers acknowledge the theoretical contribution, which goes beyond the standard i.i.d. assumption. That said, several issues should be addressed to strengthen the manuscript:

•	Motivation and experiments. The paper does not present a compelling real-world motivation; the rebuttal offered only a toy example of Markovian data. The experimental results in the reubttal are limited to classical SGD. Please compare against algorithms designed for dependent/Markovian data from the literature.
•	Novelty and related work. The algorithmic novelty appears modest (MaC-SPIDER is largely an adaptation of SPIDER), and the analysis relies on a few concentration results for Markov chains. A related-work discussion on this topic should be included, for example, to include established concentration results for MCs (e.g., Paulin, 2015).

•	Unknown parameters. The method assumes knowledge of the mixing time and the minimal visitation probability. The rebuttal’s asymptotic argument is insufficient from a sample-complexity standpoint (just mentioning an asymptotic argument of convergence  is not enough when considering sample complexity). Please discuss how much samples do we need to estimate these quantities (see e.g., Wolfer and Kontorovich papers) and explain what would be the impact on the results.

•	Presentation and correctness. The exposition can be significantly improved; e.g., the definition of mixing time is currently incorrect and should be fixed.